# MODEL-AWARE TOKENIZER TRANSFER

## ABSTRACT

Large Language Models (LLMs) are trained to support an increasing number of languages, yet their predefined tokenizers remain a bottleneck for adapting models to lower-resource or distinct-script languages. Existing tokenizer transfer methods typically rely on semantic heuristics to initialize new embeddings, ignoring higher-layer model dynamics and limiting transfer quality. We propose Model-Aware Tokenizer Transfer (MATT), a method that incorporates model internals into the tokenizer transfer process. MATT introduces an Attention Influence Modeling (AIM) objective that distills inter-token communication patterns from a source model into a target model with a new tokenizer, providing an efficient warm-up before standard language modeling. Unlike approaches that focus solely on embedding similarity, MATT leverages attention behavior to guide embedding initialization and adaptation. Experiments across diverse linguistic settings show that MATT recovers a large fraction of the original model's performance within a few GPU hours, outperforming heuristic baselines. These results demonstrate that incorporating model-level signals offers a practical and effective path toward robust tokenizer transfer in multilingual LLMs.

## 1 INTRODUCTION

Recent advances in large language models (LLMs) have shifted attention from training monolingual models (Jiang et al., 2023; Touvron et al., 2023) to covering an increasing number of languages (Grattafiori et al., 2024; Team et al., 2025). Such multilingual models have become valuable tools for researchers and practitioners working with lower-resource languages. They can be used directly for downstream tasks, help translate English datasets into the target language (Rybak, 2023), or act as a robust baseline for further adaptation (Ociepa et al., 2024). Our work focuses on the last scenario: adapting an existing LLM to a new language.

A major practical challenge in this setting is that every pretrained model is tied to a fixed tokenizer. Alternative architectures that avoid a predefined vocabulary, such as the Byte-Latent Transformer (Pagnoni et al., 2025) or H-Net (Hwang et al., 2025), are still in the experimental stage and not yet widely adopted. Tokenizers for multilingual models are usually trained to cover many scripts at once and inevitably favor high-resource languages. As a result, lower-resource languages, especially those with distinct alphabets such as Georgian, often receive a very limited share of the vocabulary. This mismatch leads not only to lower accuracy (Ali et al., 2024; Tamang & Bora, 2024), but also to slower processing and inference, which are vital for the end users.

One practical way to mitigate this problem is tokenizer transfer: replacing the original tokenizer of a pretrained model with a new one tailored to the target language and retraining the input and output embeddings (de Vries & Nissim, 2020). Even models not explicitly trained for multilinguality usually contain some cross-lingual knowledge thanks to shared alphabets or accidental language contamination (Blevins & Zettlemoyer, 2022). Consequently, if we can initialize the new embeddings well, much of the original performance can be recovered and used as a strong starting point for continual pretraining. At the same time, we should not expect this process to introduce entirely new linguistic knowledge, since several studies show that most of the model's knowledge is stored in the feed-forward layers (Dai et al., 2022; Geva et al., 2021; Nichani et al., 2024).

Most existing tokenizer-transfer methods focus almost exclusively on the embedding layer. They construct new embeddings as linear combinations of the original ones, differing mainly in how the combination weights are computed (Minixhofer et al., 2022; Dobler & de Melo, 2023; Remy et al.,

2023; 2024; Li et al., 2025). By ignoring the higher layers, these approaches overlook how the model actually processes tokens. More recent work, such as Zero-Shot Tokenizer Transfer by Minixhofer et al. (2024), leverages the full model by training a hypernetwork with a language modeling objective to predict embeddings. While effective, this strategy is computationally demanding because language modeling requires full forward and backward passes through the model.

To overcome these limitations, we introduce **Model-Aware Tokenizer Transfer (MATT)**, a method that leverages the internal behavior of the pretrained model rather than relying only on surface semantics. At the core of MATT is **Attention Influence Modeling (AIM)** objective.

AIM encourages the model with the new tokenizer to reproduce the inter-token interactions of the original model's attention layers. In effect, the original model acts as a teacher, while the model with the new tokenizer serves as a student that learns to match its attention patterns. This procedure distills structural knowledge about token relationships directly from the teacher, providing a richer and more informative initialization than relying on an embedding layer alone.

MATT is orthogonal to existing heuristics based on semantic similarity and can be combined with them. It acts as an efficient warm-up stage before conventional language model pretraining, reducing the cost of adaptation while preserving model quality.

We evaluate MATT by transferring the tokenizers of Gemma 3 (Team et al., 2025) and Qwen 3 (Team, 2025) models to extended versions that increase compression and expand coverage for several languages, including English, German, Japanese, Arabic, Swahili, and Ukrainian. Across multiple settings, MATT consistently recovers a substantial portion of the original model's performance on both generative and discriminative tasks, while requiring only a few GPU hours and outperforming heuristic-based transfer methods.

Our contributions can be summarized as follows:

- Attention Influence Modeling (AIM): a novel distillation objective that aligns the attention dynamics of two models with different tokenizers.

- Model-Aware Tokenizer Transfer (MATT): an efficient tokenizer-transfer method that exploits model dynamics instead of relying solely on semantic relationships, achieving state-of-the-art results with substantially lower computational cost than language modeling objectives.

- Comprehensive evaluation: experiments across multiple languages and models demonstrating the effectiveness and efficiency of MATT.

## 2 RELATED WORK

**Large Language Models and Vocabulary Size**  Large Language Models are becoming increasingly multilingual. Early open-source models focused almost exclusively on English (Jiang et al., 2023; Touvron et al., 2023; Almazrouei et al., 2023), but most recent releases include at least several languages and offer partial support for many more. This shift toward multilinguality has changed how researchers choose vocabulary size.

Studies show that larger vocabularies can improve model quality (Takase et al., 2025; Liang et al., 2023), but they also slow training and inference. As a result, most current foundation models use vocabularies of about 100 to 250 thousand tokens, with strongly multilingual models leaning toward the upper end. This sweet spot, first popularized by XLM-RoBERTa (Conneau et al., 2020), continues in more recent models such as Gemma (Team et al., 2025), Aya Expanse (Dang et al., 2024), and even GPT-5[1]. Going beyond this range rarely pays off: performance gains are small, and efficiency drops sharply. As a result, tokenizers cannot achieve an optimal compression rate for every language, creating a need for techniques that allow efficient transfer of tokenizers to specific languages or domains without requiring very large vocabularies.

**Heuristics-Based Embedding Initialization Methods**  When transferring a tokenizer to a new language or domain, the main challenge is initializing embeddings for tokens that did not exist in

---

[1]https://github.com/openai/tiktoken

the original model. Early work on tokenizer transfer (Artetxe et al., 2020; Gogoulou et al., 2022; de Vries & Nissim, 2020) focused on proving that transfer was possible, so embedding initialization received little attention. Simple strategies were used, including random initialization, taking the mean of existing embeddings, sampling from their distribution, copying the embedding of a random token, or using token frequency as a guide.

Later research began to exploit semantic relationships between tokens. WECHSEL (Minixhofer et al., 2022) was an influential step: it trained FastText (Bojanowski et al., 2017) embeddings for the source and target languages and used a translation vocabulary to identify the closest source tokens for each new token. New embeddings were then initialized as weighted averages of these source embeddings. Several methods followed a similar direction. OFA (Liu et al., 2024) and Tik-to-Tok (Remy et al., 2023) refined the idea of using cross-lingual similarities, while Transtokenization (Remy et al., 2024) created its own token-level translation dictionary with FastAlign (Dyer et al., 2013). Hyper-OFA (Özeren et al., 2025) went further by training a hypernetwork to map tokens from an external multilingual space into the model's embedding space, avoiding the need for simplistic linear combinations. TokAlign (Li et al., 2025) took a co-occurrence perspective, training two GloVe (Pennington et al., 2014) models on the same corpus to learn a one-to-one alignment matrix between tokens.

As LLMs became more multilingual, overlap between source and target vocabularies became an important resource. FOCUS (Dobler & de Melo, 2023) trains a FastText model on text tokenized with the target vocabulary, then initializes new embeddings as similarity-weighted averages of overlapping tokens. CLP Transfer (Ostendorff & Rehm, 2023) takes advantage of topological similarities of the latent space across model sizes within the same family: embeddings are first trained on a smaller related model and then aligned to the target model by measuring similarities with overlapping tokens.

**Beyond Heuristics**   While heuristics provide a practical starting point, they have limitations. An alternative is to train new embeddings directly by continuing language modeling with all other parameters frozen (de Vries & Nissim, 2020), but this is computationally costly.

Mini-Model Adaptation (Marchisio et al., 2023) reduces the cost by using only a subset of layers and training the embeddings on a language modeling task. Other work (Chen et al., 2023) shows that periodically resetting embeddings during pretraining makes models more robust to them, reducing the effort needed to learn new tokens afterwards.

Another approach by Minixhofer et al. (2024) trains a universal hypernetwork for a given language model by sampling tokenizers from a diverse distribution during the language modeling stage. Once the hypernetwork is trained, we can initialize embeddings for various tokenizers effortlessly, achieving a solid baseline for further continual pretraining. However, training such a hypernetwork is a compute-heavy task, requiring forward and backward passes through the whole model in every step to update the hypernetwork weights, limiting its practicality in settings where we already have a defined target tokenizer and the trained hypernetwork is not available beforehand.

## 3   METHOD

### 3.1   INTUITION

Large Language Models generate text one token at a time. Decoder-only transformers, which form the backbone of most modern LLMs, follow the following steps: the embedding of the most recently generated token is passed through a stack of attention and feed-forward layers, and finally projected by the LM head to produce a probability distribution over the next token.

Assuming the input embedding of the last token is correct, the feed-forward layers will not damage its representation. The main source of potential distortion lies in the attention layers, where each token interacts with the context. Changing the tokenizer introduces new tokens into the context, altering these interactions and thus the internal representations that drive next-token prediction. Our goal is to train a model using a new tokenizer so that, despite these changes, its attention layers produce output embeddings similar to those generated by the original tokenizer.

## 3.2 PREREQUISITES

Consider an input string $s$ and a tokenization function $T$, which produces a token sequence $T(s) = (t_1, t_2, \ldots, t_n)$ of length $n$.

In each attention layer[2], the inputs are the query ($Q$), key ($K$), and value ($V$) state matrices, producing the output state matrix ($O$). Each of these can be seen as a collection of vector states for every token $t_i$:

$$
Q = \begin{bmatrix} q_1 \\ q_2 \\ \cdots \\ q_n \end{bmatrix}_{n \times h} \quad
K = \begin{bmatrix} k_1 \\ k_2 \\ \cdots \\ k_n \end{bmatrix}_{n \times h} \quad
V = \begin{bmatrix} v_1 \\ v_2 \\ \cdots \\ v_n \end{bmatrix}_{n \times h} \quad
O = \begin{bmatrix} o_1 \\ o_2 \\ \cdots \\ o_n \end{bmatrix}_{n \times h} ,
$$

where $h$ is the hidden size.

Attention is computed as:

$$
O = \text{Attention}(Q, K, V) = \text{softmax}\left( \frac{QK^T}{\sqrt{d_k}} \right) V = AV,
$$

where $A$ is the attention matrix of shape $n \times n$, that contains weights with which the value states are aggregated into the output state.

We can break down the final matrix multiplication $AV$ into a chain of value states ($V$) averages for each token, weighted by the attention matrix $A$. The output state for the token $t_i$ would then look the following way:

$$
o_i = \text{softmax}\left( \frac{q_i K^T}{\sqrt{d_k}} \right) V = A_{i,:} V = \sum_{j=1}^{n} A_{i,j} v_j = \sum_{j=1}^{n} v_{i,j}^*,
$$

where $v_{i,j}^* = A_{i,j} v_j$ is a weighted value state for the token $t_j$ given the query token $t_i$.

## 3.3 SEGMENT-LEVEL INTERPRETATION OF ATTENTION

To compare attention outputs across different tokenizers, we introduce a segmentation function $S$ that splits the input string $s$ into segments $(s_1, s_2, \ldots, s_m)$ while respecting a set of tokenization functions $\mathcal{T}$:

$$
S(s; \mathcal{T}) = (s_1, s_2, \ldots, s_m), \text{ such that}
$$
$$
\forall T \in \mathcal{T}: T(s_1) \circ T(s_2) \circ \cdots \circ T(s_m) = T(s),
$$

where $\circ$ is a concatenation operator. This ensures that no segment boundary lies within any token produced by any tokenization function in $\mathcal{T}$.

The most intuitive approach is a function that splits the input string into words, and the rest of the section is explained in relation to this function. However, for some languages, word segmentation can be ambiguous; thus, in practice, we define our segmentation function to always choose segments of minimal length that still satisfy the above condition (see Appendix A for the algorithm).

Given $S$, we define weighted value states for a segment $s_k$ with respect to a query token $t_i$:

$$
\mathfrak{s}_{i,k} = \sum_{\{j \, : \, t_j \in T(s_k)\}} v_{i,j}^*
$$

The output state for token $t_i$ can then be expressed as a sum over segments:

---

[2]While we proceed with a single-head definition, it is directly applicable to multi-head, multi-query, or grouped-query attention variants.

$$\boldsymbol{o}_i = \sum_{j=1}^{n} \boldsymbol{v}_{i,j}^* = \sum_{j=1}^{m} \mathfrak{s}_{i,j}$$

To move from token-level to segment-level interpretation, we replace individual query tokens with segment representations. Since the output state of each token is designed to predict the next token, it is natural to require that the output state of a segment should similarly carry enough information to predict the next segment. Because the language modeling head still operates at the token level, we approximate "predicting the next segment" by predicting the first token of that next segment.

Consider a segment $s_i$ whose tokens are $T(s_i) = (t_a, t_{a+1}, \ldots, t_b)$. The final token $t_b$ produces the output state used to generate the next token $t_{b+1}$, which begins the following segment $s_{i+1}$. We therefore define a function $\ell_T$ that maps a segment index to the index of its last token:

$$\ell_T(i) = b$$

The query state of segment $s_i$ is set equal to the query state of its last token:

$$\mathbf{q}_i = \boldsymbol{q}_{\ell_T(i)} = \boldsymbol{q}_b,$$

and the output state of the segment is computed from this query state:

$$\mathbf{o}_i = \mathrm{softmax}\left(\frac{\mathbf{q}_i \boldsymbol{K}^T}{\sqrt{d_k}}\right)\boldsymbol{V} = \mathrm{softmax}\left(\frac{\boldsymbol{q}_{\ell_T(i)} \boldsymbol{K}^T}{\sqrt{d_k}}\right)\boldsymbol{V} = \boldsymbol{o}_{\ell_T(i)} = \sum_{j=1}^{m} \mathfrak{s}_{\ell_T(i),j}$$

### 3.4 ATTENTION INFLUENCE MODELING

As described in the Section 3.1, our goal is to train the model with a new tokenizer $T'$ so that its output states match those of the original model with tokenizer $T$.

Since we can enforce a common segmentation function $S$, we approximate this by requiring the new model to produce the same segment-level outputs $\mathbf{o}_i'$ as the old ones – $\mathbf{o}_i$. A more detailed objective also matches the weighted value states $\mathfrak{s}_{\ell_T(i),j}$ and $\mathfrak{s}_{\ell_{T'}(i),j}'$ of every segment $s_j$ for each query segment $s_i$, with the causal constraint $j \leq i$.

Given the above, we define the **Attention Influence Modeling** objectives (normal and simplified):

$$\mathcal{L}_{AIM} = \frac{2}{m(m+1)} \sum_{i=1}^{m} \sum_{j=1}^{i} \mathcal{L}^*(\mathfrak{s}_{\ell_T(i),j}, \mathfrak{s}_{\ell_{T'}(i),j}'),$$

$$\mathcal{L}_{AIM^*} = \frac{1}{m} \sum_{i=1}^{m} \mathcal{L}^*(\mathbf{o}_i, \mathbf{o}_i'),$$

where $\mathcal{L}^*(\boldsymbol{x}, \boldsymbol{y})$ can be any loss function, that brings $\boldsymbol{x}$ and $\boldsymbol{y}$ closer. In Section 4, we experiment with MSE and Cosine Embedding losses.

Figure 1 illustrates an example of applying AIM to the text *CH4 – formula for methane*. In this case, we use a word segmentation function together with different tokenization functions for a given query state $\mathbf{q}_5$, where the segment $s_5$ corresponds to _methane.

Figure 2 presents the attention alignment matrix for the same text, where the weighted value states $\boldsymbol{v}_{i,j}^*$ are grouped into segments. These segment-level representations are then matched and optimized to be equal under the $\mathcal{L}_{AIM}$ loss.

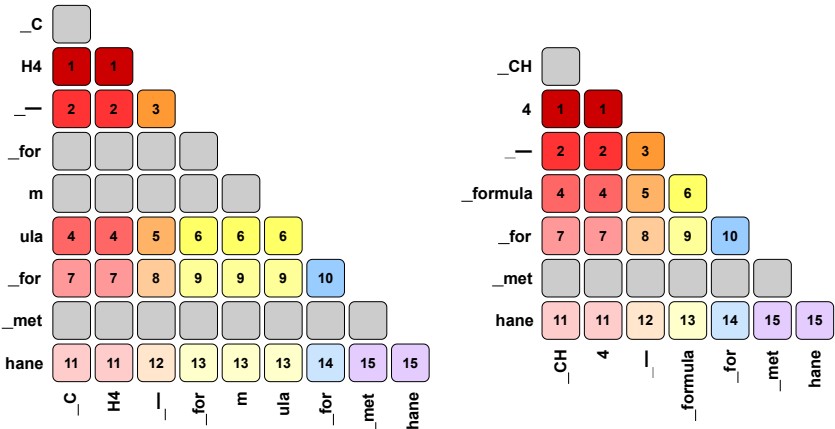

Figure 1: Attention Influence Modeling (AIM) objective with word segmentation. For each input, the weighted value vectors $v_{i,j}^*$ of the original tokens $t_j$ are aggregated into segment-level vectors $\mathfrak{s}_{i,k}$ according to a chosen word-segmentation function. The model trained with the new tokenizer produces its own segment representations $\mathfrak{s}_{i,k}'$. The AIM objective encourages these new segment representations to stay close to the segment representations $\mathfrak{s}_{i,k}$ computed from the model using the old tokenizer. All this happens with respect to the query state $q_5$ of the 5th segment (_methane), which is equal to the query state of its last token – hane.

Figure 2: Token-level attention alignment between teacher and student models. The left matrix shows the weighted value states of the teacher model using the original tokenizer $T$, and the right matrix shows those of the student model using the new tokenizer $T'$. Each square represents the weighted value state $v_{i,j}^*$ of $t_j$ for query token $t_i$ ($i$ for rows, $j$ for columns). Numbers (or matching colors) within a matrix identify tokens that are aggregated into the same segment-level state $\mathfrak{s}_{i,k}$. Numbers (or matching colors) across the two matrices indicate corresponding pairs $\mathfrak{s}_{\ell_T(i),j}$ and $\mathfrak{s}_{\ell_{T'}(i),j}'$ used in the loss $\mathcal{L}^*$ to align the teacher and the student attention representations.

## 3.5 TECHNICAL DETAILS

During training, the model with the old tokenizer $T$ is kept frozen. The model with the new tokenizer $T'$ has all layers frozen except the input embeddings. As a small modification to the basic training setup, we partially freeze the embedding matrix: tokens that are shared between the old and new tokenizers are initialized from the original model and kept fixed, while only the embeddings of new, non-overlapping tokens are updated during training.

To speed up convergence, we initialize new embeddings using FOCUS (Dobler & de Melo, 2023). We train with AdamW (Loshchilov & Hutter, 2017), a constant learning rate of $1 \times 10^{-4}$, and no weight decay. However, it should be noted that we have not performed extensive hyperparameter tuning, so using learning rate scheduling, adapting the learning rate, weight decay, and other hyperparameters may yield significantly better results.

MATT offers a key advantage over standard language modeling with frozen non-embedding parameters: greater efficiency. Since AIM is defined at the attention-layer level, we can decide how much

of the model to include in the tokenizer transfer by selecting the layer depth at which AIM is applied. Specifically, by choosing a value of $n$, we take only the first $n$ layers into account. This allows us to balance efficiency and performance.

We ablate the choice of MATT target layer in Appendix C and observe that using higher layers improves performance until roughly the first quarter of the model, aligning with the formation of coherent word-level representations as tokens are detokenized in early and middle layers, a behavior noted in prior work as detokenization (Kaplan et al., 2024). Targeting the final layers causes slight degradation, consistent with trends reported in Token Distillation Dobler et al. (2025), while middle layers show a performance plateau. Because later target layers increase training time and memory linearly (validated in our appendix ablations), we select one of the earliest layers within this plateau to balance strong performance with minimal resource cost.

Since only input embeddings are trained, tied input–output embeddings are advantageous, as the tuned input embeddings can be reused in the LM head. Models without tied embeddings still benefit from input tuning, but to a significantly lesser extent; handling untied settings is left for future work.

## 4 EXPERIMENTS

We conducted a series of experiments across different languages, model families, and scales to evaluate the effectiveness of the MATT method compared to existing heuristic- and optimization-based approaches. In each experiment, we first trained a tokenizer with a higher compression rate than the original one, merged it with the base tokenizer, and then applied tokenizer transfer to the extended vocabulary. We have chosen Ukrainian as a language with Cyrillic alphabet that is not well represented in the dictionaries of the major LLMs and on the other hand that is included in multiple benchmarks, allowing for the inspection of the method's performance in different scenarios. For the multilingual setting we have chosen five typologically diverse languages – English, German, Japanese, Arabic, and Swahili, which vary in resource availability, writing system, and language family.

Additional experiments, including convergence speed tests (Appendix B) and ablation studies (Appendix C), are presented to complement the main results.

### 4.1 MAIN RESULTS

Our primary evaluation uses the Gemma 3 12B PT model (Team et al., 2025). We replaced its default tokenizer with an extended version that improves Ukrainian coverage, raising the compression rate from 2.98 to 4.44. This increase translates to an almost 50% speedup during inference. We compare the following methods:

- **WECHSEL** – transfer using the English–Ukrainian vocabulary from the official implementation[3].
- **Transtokenizers** – token alignment via FastAlign using parallel corpora (OpenSubtitles (Lison & Tiedemann, 2016) and NLLB (NLLB Team, 2022)) and the official `transtokenizers`[4] toolkit.
- **TokAlign** – GloVe embeddings trained on 2 million Ukrainian documents (approximately 1.86 billion Gemma tokens) from the Kobza corpus (Haltiuk & Smywiński-Pohl, 2025), used to create a one-to-one alignment matrix with the official implementation[5].
- **FOCUS** – FastText embeddings trained on the same data as TokAlign, with initialization performed via the `deepfocus`[6] package.
- **NTP** – initialized with one of the above methods, and trained using the Next Token Prediction (NTP) objective with non-embedding layers frozen. We compare several versions of this baseline corresponding to 50%, 100%, and 150% of the training budget dedicated to MATT.

---

[3]https://github.com/CPJKU/wechsel
[4]https://github.com/LAGoM-NLP/transtokenizer
[5]https://github.com/ZNLP/TokAlign
[6]https://github.com/konstantinjdobler/focus

- **MATT** – initialized with FOCUS embeddings and trained on around 240 million Ukrainian tokens from Kobza using the AIM objective with MSE loss on the 12th layer out of 34, original embeddings are frozen, and all other hyperparameters remain unchanged (see Section 3.5).

We evaluate performance on Belebele (Bandarkar et al., 2024), Global MMLU (Singh et al., 2025), Long FLORES (Paniv, 2025), a modification of FLORES (NLLB Team, 2022; Goyal et al., 2021; Guzmán et al., 2019), which elevates the sentence-level translation to document-level by aggregating data points from the same sources, WMT24++ (Deutsch et al., 2025), and XL-SUM (Hasan et al., 2021). We only evaluate the translations from English to Ukrainian with a specific intent to validate the model's performance on a generative task in the target language. Evaluation is performed with the `lm-evaluation-harness` framework (Gao et al., 2024) with a 3-shot prompt.

Table 1: Performance of Gemma 3 12B PT model with different tokenizer transfer methods on Belebele and Global MMLU (accuracy, %), Long FLORES, WMT, and XL-SUM (BLEU). The "Avg Disc" column reports the average of Belebele and Global MMLU scores, as well as "Avg Gen" – of Long FLORES, WMT, and XL-Sum.

| Model | Training Time | Belebele | Global MMLU | Long FLORES | WMT | XL-Sum | Avg Disc | Avg Gen |
|---|---|---|---|---|---|---|---|---|
| Gemma 3 12B PT | - | 89.33 | 67.03 | 14.36 | 3.52 | 6.52 | 78.18 | 8.13 |
| *Heuristics* | | | | | | | | |
| WECHSEL | - | 22.67 | 24.61 | 0.00 | 0.00 | 0.00 | 23.64 | 0.00 |
| Transtokenizers | - | 61.89 | 46.03 | 0.04 | 0.09 | 0.02 | 53.96 | 0.05 |
| TokAlign | - | 31.44 | 32.98 | 0.00 | 0.00 | 0.01 | 32.21 | 0.00 |
| FOCUS | - | 48.78 | 37.14 | 1.01 | 0.88 | 0.20 | 42.96 | 0.70 |
| *Optimization Based* | | | | | | | | |
| Transtokenizers w/ NTP | 3h 30m | 82.44 | 59.02 | 3.64 | 0.88 | 4.06 | 70.73 | 2.86 |
| Transtokenizers w/ NTP | 7h 00m | 85.22 | 59.83 | 4.63 | 0.95 | 4.80 | 72.53 | 3.46 |
| Transtokenizers w/ NTP | 10h 30m | 85.67 | 59.38 | 5.13 | 0.96 | 4.80 | 72.53 | 3.63 |
| FOCUS w/ NTP | 3h 30m | 85.44 | 57.38 | 3.51 | 2.13 | 4.32 | 71.41 | 3.32 |
| FOCUS w/ NTP | 7h 00m | 87.00 | 60.55 | 4.32 | 2.51 | 5.04 | 73.78 | 3.96 |
| FOCUS w/ NTP | 10h 30m | 87.44 | 60.57 | 4.34 | 2.60 | 5.16 | 74.01 | 4.03 |
| MATT | 7h 00m | **89.56** | **64.98** | **8.70** | **4.71** | **5.95** | **77.27** | **6.45** |

Table 1 shows a clear advantage of MATT over all other methods. While heuristic-based approaches such as FOCUS and Transtokenizers can regain up to about 70% of the original model's accuracy on discriminative tasks, they reach no more than about 9% of the original generative performance.

Optimization-based approaches show substantially stronger recovery. Among them, NTP yields consistent results across different heuristic initializations. In its best configuration, NTP recovers nearly 95% of the original model's discriminative performance, though only about 50% of its generative capabilities. In contrast, MATT restores nearly 80% of the original generative performance while maintaining accuracy on discriminative tasks close to the unmodified model. These results demonstrate the superiority of the model-aware approach to tokenizer transfer, particularly given the extremely low computational costs required.

The most notable observation is the minimal improvement of NTP when comparing 100% and 150% compute budgets (7 hours and 10.5 hours of training time respectively). NTP training rapidly saturates, and MATT-level performance does not appear to be attainable within a reasonable budget. We therefore stop further training due to our limited computational resources.

MATT also saturates quickly (see Appendix B), but at a substantially higher performance level. We hypothesize that further gains are unlikely under embedding-only training and will require unfreezing the model's layers. This is largely driven by the scale of newly introduced tokens (over 80,000), which meaningfully alters model dynamics and would likely benefit from full fine-tuning. Due to computational constraints, we have conducted only limited experiments with continual pretraining after MATT that provide early support for this hypothesis. A systematic comparison of embedding-only adaptation and full model fine-tuning remains a direction for future work.

## 4.2 MULTILINGUAL RESULTS

In the multilingual setting, we experiment with Gemma 3 4B PT and Qwen 3 0.6B (Team, 2025). As shown in Table 2, the extended tokenizers consistently improve compression rates across all languages, including modest gains for English, which directly translates to their processing and generation speed, since a considerably smaller number of tokens is required to represent the same text.

Table 2: Comparison of original and extended tokenizers. Compression rate is the average number of characters represented by a single token (higher is better).

| Tokenizer | Vocabulary Size | Compression Rate | | | | |
|---|---|---|---|---|---|---|
| | | ar | de | en | ja | sw |
| **Gemma** | | | | | | |
| original | 262,145 | 2.8457 | 3.9734 | 4.3187 | 1.6846 | 2.9802 |
| extended | 387,980 | **3.9122** | **4.4997** | **4.3383** | **2.1267** | **4.2518** |
| **Qwen** | | | | | | |
| original | 151,669 | 2.5982 | 3.4737 | 4.3599 | 1.4852 | 2.5788 |
| extended | 298,833 | **3.9221** | **4.4886** | **4.4233** | **2.2867** | **4.2322** |

The transfer methods remain the same as in Section 4.1, except the training data is now drawn from HPLT 2.0 Cleaned (Burchell et al., 2025). FOCUS uses 2 million documents (approximately 500 million tokens per language), and MATT – only about 50 million tokens per language. We also experiment with an AIM* objective, Cosine Embedding loss, and training without freezing the original embeddings. Performance is reported on Belebele, MMMLU (Hendrycks et al., 2020), and Global MMLU. We additionally record the time and memory required to tune embeddings on a single H100 GPU.

Table 3: Benchmark results for transferring original tokenizers to their extended versions across five languages (Arabic, German, English, Japanese, Swahili). For the proposed MATT method, peak VRAM usage and processing time required for the tokenizer transfer are also reported.

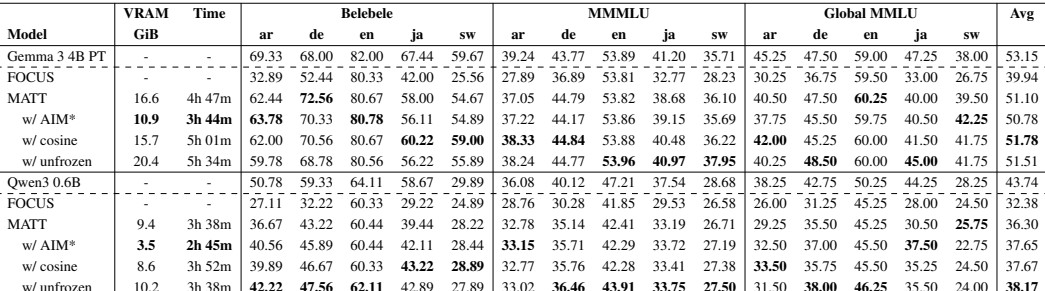

| Model | VRAM GiB | Time | Belebele ar | de | en | ja | sw | MMMLU ar | de | en | ja | sw | Global MMLU ar | de | en | ja | sw | Avg |
|---|---|---|---|---|---|---|---|---|---|---|---|---|---|---|---|---|---|---|
| Gemma 3 4B PT | - | - | 69.33 | 68.00 | 82.00 | 67.44 | 59.67 | 39.24 | 43.77 | 53.89 | 41.20 | 35.71 | 45.25 | 47.50 | 59.00 | 47.25 | 38.00 | 53.15 |
| FOCUS | - | - | 32.89 | 52.44 | 80.33 | 42.00 | 25.56 | 27.89 | 36.89 | 53.81 | 32.77 | 28.23 | 30.25 | 36.75 | 59.50 | 33.00 | 26.75 | 39.94 |
| MATT | 16.6 | 4h 47m | 62.44 | **72.56** | 80.67 | 58.00 | 54.67 | 37.05 | 44.79 | 53.82 | 38.68 | 36.10 | 40.50 | 47.50 | **60.25** | 40.00 | 39.50 | 51.10 |
| w/ AIM* | **10.9** | **3h 44m** | **63.78** | 70.33 | **80.78** | 56.11 | 54.89 | 37.22 | 44.17 | 53.86 | 39.15 | 35.69 | 37.75 | 45.50 | 59.75 | 40.50 | **42.25** | 50.78 |
| w/ cosine | 15.7 | 5h 01m | 62.00 | 70.56 | 80.67 | **60.22** | **59.00** | **38.33** | **44.84** | 53.88 | 40.48 | 36.22 | **42.00** | 45.25 | 60.00 | 41.50 | 41.75 | **51.78** |
| w/ unfrozen | 20.4 | 5h 34m | 59.78 | 68.78 | 80.56 | 56.22 | 55.89 | 38.24 | 44.77 | **53.96** | **40.97** | **37.95** | 40.25 | **48.50** | 60.00 | **45.00** | 41.75 | 51.51 |
| Qwen3 0.6B | - | - | 50.78 | 59.33 | 64.11 | 58.67 | 29.89 | 36.08 | 40.12 | 47.21 | 37.54 | 28.68 | 38.25 | 42.75 | 50.25 | 44.25 | 28.25 | 43.74 |
| FOCUS | - | - | 27.11 | 32.22 | 60.33 | 29.22 | 24.89 | 28.76 | 30.28 | 41.85 | 29.53 | 26.58 | 26.00 | 31.25 | 45.25 | 28.00 | 24.50 | 32.38 |
| MATT | 9.4 | 3h 38m | 36.67 | 43.22 | 60.44 | 39.44 | 28.22 | 32.78 | 35.14 | 42.41 | 33.19 | 26.71 | 29.25 | 35.50 | 45.25 | 30.50 | **25.75** | 36.30 |
| w/ AIM* | **3.5** | **2h 45m** | 40.56 | 45.89 | 60.44 | 42.11 | 28.44 | **33.15** | 35.71 | 42.29 | 33.72 | 27.19 | 32.50 | 37.00 | 45.50 | **37.50** | 22.75 | 37.65 |
| w/ cosine | 8.6 | 3h 52m | 39.89 | 46.67 | 60.33 | **43.22** | **28.89** | 32.77 | 35.76 | 42.28 | 33.41 | 27.38 | **33.50** | 35.75 | 45.50 | 35.25 | 24.50 | 37.67 |
| w/ unfrozen | 10.2 | 3h 38m | **42.22** | **47.56** | **62.11** | 42.89 | 27.89 | 33.02 | **36.46** | **43.91** | **33.75** | **27.50** | 31.50 | **38.00** | **46.25** | 35.50 | 24.00 | **38.17** |

Table 3 shows that MATT substantially narrows the performance gap between a freshly initialized model and the original, recovering most of the accuracy and occasionally surpassing the original.

Regarding the training objectives, the Cosine Embedding loss generally provides strong performance across the discriminative benchmarks reported in Table 3. However, we note from preliminary experiments that Mean Squared Error (MSE) tends to be more beneficial for generation-heavy tasks, and therefore stands as our default choice.

A critical observation regarding the effectiveness of MATT is its behavior when the base model exhibits near-random performance. As a self-distillation method, MATT is designed to recover the original model's capabilities rather than induce cross-lingual transfer for unseen languages. This is evident in the results for Qwen3 0.6B on Swahili. Although the Qwen 3 family technically supports Swahili, the 0.6B model shows minimal difference from random baselines, likely due to limited model capacity and a small share in pretraining data mixture. Despite this unfavorable setting,

MATT successfully recovers the majority of the original model's performance, effectively leveraging a weak original signal.

We also find that the choice of freezing embeddings has an impact on performance depending on the adaptation scope. Unfreezing all embeddings yields the best results in this multilingual setting. We attribute this to the need for greater model elasticity when adapting to five diverse languages simultaneously. In contrast, freezing the original embeddings, as done in Section 4.1, remains the practical choice for single-language adaptation, where the priority is often to preserve performance on the original language (e.g., English) while extending coverage to another target language.

Based on these findings, we recommend distinct default configurations: for single-language adaptation with limited vocabulary extension, a combination of MSE loss, the standard AIM objective, and frozen original embeddings is optimal. For multilingual adaptation, unfreezing the original embeddings is preferable to accommodate broader semantic shifts. The AIM* variant offers a good compromise, reducing memory and runtime while only slightly lowering accuracy. Further VRAM savings are possible with a custom kernel for AIM computation.

Finally, we observe a disparity in robustness relative to model size. Larger models appear more resilient to embedding initialization shocks. This is illustrated by the FOCUS initialization results on English: while Gemma 3 4B PT maintains high stability, Qwen3 0.6B suffers a notable performance drop even when only a few new tokens are introduced. This suggests that larger parameter counts may provide a buffer against the perturbations introduced during tokenizer transfer. However, this behavior may also be intrinsic to the specific model families, and thus requires further experimentation.

## 5 CONCLUSION

In this work, we introduced MATT, a model-aware method for tokenizer transfer that leverages the internal dynamics of LLMs. We applied MATT to extend the tokenizers of Gemma 3 and Qwen 3 models across multiple languages and settings, demonstrating that it consistently recovers a large portion of the original model's capabilities while requiring only a few GPU hours of training. Unlike heuristic-based methods that rely solely on the embedding layer, MATT refines token representations with direct feedback from the model, thanks to the novel Attention Influence Modeling (AIM) objective, allowing it to bridge the performance gap caused by tokenizer changes more effectively.

Our experiments highlight this advantage most clearly in the transfer of the 12 billion-parameter Gemma 3 model to Ukrainian. With the extended tokenizer introducing over 80,000 new tokens, MATT achieves an average score of 77.27 out of the original 78.18 on the discriminative tasks and 6.45 out of 8.13 on the generative ones, outperforming both heuristic and optimization-based baselines. This substantial improvement underscores the value of incorporating model dynamics into tokenizer transfer and shows that high performance can be retained at a fraction of the computational cost typically required for NTP training.

### LIMITATIONS

The first limitation lies in the fact that MATT relies on tied input and output embeddings to fully realize its advantages. We outline possible strategies to relax this requirement in the Appendix D.

Second, we do not perform continual pretraining with all weights unfrozen due to computational constraints, and instead evaluate only models with initialized or trained embedding layers. This is sufficient to compare MATT with existing baselines, whose primary goal is to provide a strong starting point for further adaptation.

A further limitation is the need for an additional forward pass during optimization through the model using the original tokenizer to obtain targets for the AIM objective. Although this adds computational overhead, the cost remains lower than a full forward pass through the entire model, as the target layer is positioned roughly one-third of the way through the network.

Finally, we have not tested MATT on encoder-only architectures. In principle, applying it to such models would only require removing the causal constraint in the AIM definition.

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

## A    SEGMENTATION ALGORITHM

Instead of relying on word-based segmentation, we use an offset-based segmentation strategy. Designing a consistent word segmentation across tokenizers is challenging because tokenizers often differ in normalization rules, pre-tokenization steps, language coverage, etc. These differences make it difficult to ensure that segment boundaries match at the word level.

The offset-based method addresses this by operating directly on character offsets in the original text. Given two different tokenizations of the same string, along with the start and end positions of each token, the algorithm searches for all possible split positions that never cut through the middle of any token (see Figure 3).

This approach is universal: such a segmentation always exists, even if the worst case reduces to a single segment spanning the entire input. It can also lead to more precise alignments because the target tokenizer may break a word into several sub-tokens. By working with character offsets, we can introduce mid-word segment boundaries whenever they yield a better match.

For example, consider the sentence *CH4 is a formula for methane*. Suppose the original tokenizer produces the tokens _for, m, and ula for the word *formula*, while the new tokenizer produces _form and ula. A word-level strategy would force alignment at the whole-word boundary, but an offset-based method can instead match _for and m with _form, and ula with ula, which more closely respects both tokenizations.

Algorithm 1 provides detailed pseudocode for implementation.

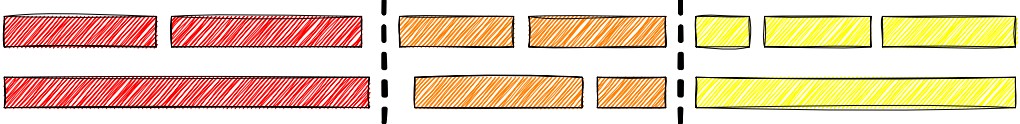

Figure 3: Offset-based segmentation algorithm visualization.

**Algorithm 1:** Offset–Based Segmentation

**Input:** teacher offsets $O_t$, student offsets $O_s$
**Output:** teacher segment ids $S_t$, student segment ids $S_s$

▷ initialize outputs and counters

1   $S_t \leftarrow [\,]$, $S_s \leftarrow [\,]$
2   $e \leftarrow -1$;   ▷ current end
3   $k \leftarrow -1$;   ▷ current segment id

▷ iterate until both queues empty

4   **while** $O_t \neq \emptyset$ *or* $O_s \neq \emptyset$ **do**

     ▷ if one side empty, label all remaining tokens with current segment id

5    **if** $O_t = \emptyset$ **then**
6      **for** *each o in $O_s$* **do**
7       append $k$ to $S_s$
8      **break**
9    **else if** $O_s = \emptyset$ **then**
10     **for** *each o in $O_t$* **do**
11      append $k$ to $S_t$
12     **break**

     ▷ peek next offsets

13   $(t_s, t_e) \leftarrow \text{peek}(O_t)$, $(s_s, s_e) \leftarrow \text{peek}(O_s)$

     ▷ continue with the same segment if overlap

14   **if** $t_s < e$ **then**
15    append $k$ to $S_t$, $\text{pop}(O_t)$
16    $e \leftarrow \max(e, t_e)$
17   **else if** $s_s < e$ **then**
18    append $k$ to $S_s$, $\text{pop}(O_s)$
19    $e \leftarrow \max(e, s_e)$

     ▷ else start a new segment

20   **else**
21    $k \leftarrow k + 1$
22    append $k$ to $S_t$ and $S_s$, $\text{pop}(O_t)$, $\text{pop}(O_s)$
23    $e \leftarrow \max(t_e, s_e)$

24   **return** $(S_t, S_s)$

## B   CONVERGENCE SPEED

We repeated the experiment with Gemma 3 4B PT described in Section 4.2, but this time we saved model checkpoints every 3,000 training steps. While the results in Table 3 were obtained after 250,000 steps, this setup allows us to observe how quickly the embeddings adapt to the new tokenizer and to evaluate whether training can be substantially shortened.

The AIM objective provides a rich learning signal for tuning the embeddings. Using a mean squared error (MSE) loss, the number of value pairs contributing to the objective is proportional to the product of the head dimensionality, the number of attention heads, the number of possible segment pairs, and the batch size. In the configuration used here – four documents per batch, each truncated to 256 tokens – this amounts to hundreds of millions of pairs at every step of the training.

Table 4 presents the results for the first eight checkpoints on the Belebele benchmark across all tested languages, while Figure 4 provides a visual view of the same trends.

The data show that more than 50% of the final performance gains can be achieved in under 10% of the total training steps, corresponding to fewer than five million tokens per language. This indicates that training time could be cut dramatically with only a minor loss in accuracy, especially if an adaptive data selection strategy is used to prioritize documents that contain a higher proportion of previously unseen tokens.

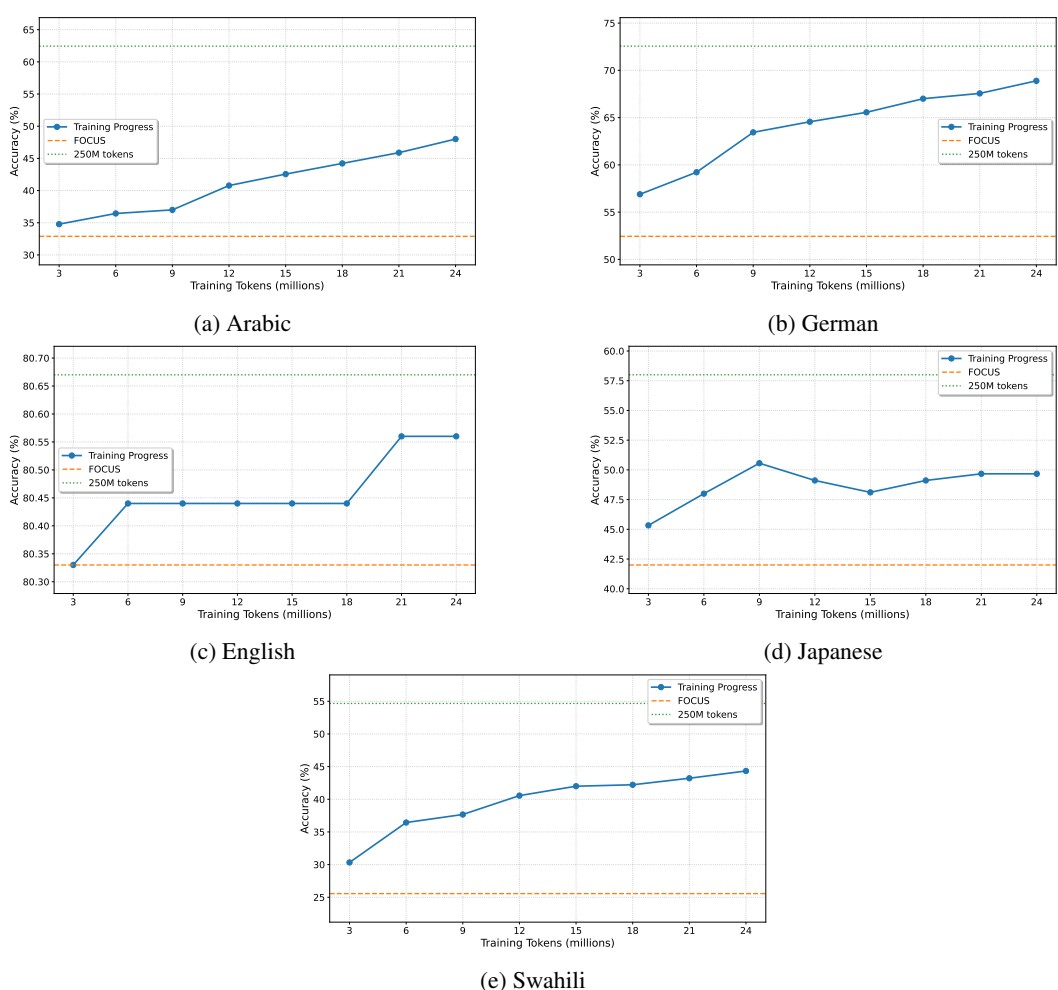

Figure 4: Accuracy on the Belebele benchmark over training tokens for five languages, with horizontal lines marking FOCUS initialization and full performance.

## C  ABLATION STUDIES

We apply the MATT method with AIM objective, transferring Gemma 3 4B PT to a Ukrainian-centric tokenizer by Bohdan Didenko (2025), which increases the compression rate by around 50%. We conduct several ablation studies to determine the optimal training configuration, evaluating performance on the Belebele and Global MMLU benchmarks in the same setting as in Section 4. We report the results in Table 5 and describe our insights below.

**AIM objective on all layers deteriorates both efficiency and performance compared to only the last one.** Since AIM is defined for a single attention layer, we can apply it many times to any subset of layers. To balance efficiency and accuracy, we compare defining the AIM objective over all layers up to a target depth $n$ against using only the final $n$-th layer. Results in Table 5 show that restricting AIM to the last layer requires much less VRAM and training time, while also delivering better downstream performance.

Table 4: Performance on the Belebele benchmark during early training of Gemma 3 4B PT with Model-Aware Tokenizer Transfer, showing rapid gains within the first 10% of steps compared to the full run.

| | | Belebele | | | | |
|---|---|---|---|---|---|---|
| **Steps #** | **Tokens #** | **ar** | **de** | **en** | **ja** | **sw** |
| 0k | 0M | 32.89 | 52.44 | 80.33 | 42.00 | 25.56 |
| 3k | 3M | 34.78 | 56.89 | 80.33 | 45.33 | 30.33 |
| 6k | 6M | 36.44 | 59.22 | 80.44 | 48.00 | 36.44 |
| 9k | 9M | 37.00 | 63.44 | 80.44 | 50.56 | 37.67 |
| 12k | 12M | 40.78 | 64.56 | 80.44 | 49.11 | 40.56 |
| 15k | 15M | 42.56 | 65.56 | 80.44 | 48.11 | 42.00 |
| 18k | 18M | 44.22 | 67.00 | 80.44 | 49.11 | 42.22 |
| 21k | 21M | 45.89 | 67.56 | 80.56 | 49.67 | 43.22 |
| 24k | 24M | 48.00 | 68.89 | 80.56 | 49.67 | 44.33 |
| 250k | 250M | 62.44 | 72.56 | 80.67 | 58.00 | 54.67 |

Table 5: Ablation studies of MATT configurations. **3** and **5** denote the number of layers used for AIM objective.

| | VRAM (GiB) | | Time | | Belebele | | Global MMLU | |
|---|---|---|---|---|---|---|---|---|
| | **3** | **5** | **3** | **5** | **3** | **5** | **3** | **5** |
| **All Layers vs. Last Layer** | | | | | | | | |
| all layers | 17.3 | 26.5 | 1h 33m | 2h 19m | 32.56 | 35.11 | 28.63 | 29.00 |
| last layer | **9.1** | **10.1** | **0h 56m** | **1h 04m** | **37.22** | **60.11** | **29.95** | **34.80** |
| **Initialization Method** | | | | | | | | |
| WECHSEL | **9.1** | **10.1** | **0h 52m** | **1h 04m** | 42.22 | 59.78 | 29.82 | 33.56 |
| FOCUS | **9.1** | **10.1** | 0h 55m | 1h 06m | 37.11 | **60.89** | 30.10 | 34.72 |
| Transtokenizers | **9.1** | **10.1** | 0h 55m | **1h 04m** | 52.44 | **60.89** | 32.43 | 34.75 |

**FOCUS and Transtokenizers perform similarly on higher layers, while WECHSEL underperforms.** Because MATT is independent of the embedding initialization method, different starting points can be tested. We compare WECHSEL, FOCUS, and Transtokenizers. FOCUS and Transtokenizers perform similarly on higher layers, while WECHSEL lags behind (see Table 5). Although Transtokenizers occasionally achieves the best scores, in other experiments, we find FOCUS to be more stable across models, and therefore make it our default choice.

Transtokenizers method may have an upper hand due to its better utilization of English embeddings, as it learns an English-Ukrainian token-level dictionary from parallel corpora and utilizes it to transfer embeddings from English tokens to their Ukrainian counterparts. Whereas FOCUS utilizes the tokens' overlap to train a FastText model over it, and although it contains English tokens as well, the FastText training corpus contains little data that encompasses both English and Ukrainian text in the same document. This could potentially limit the FOCUS to pay attention mostly to Ukrainian overlapped tokens, given the limited usability of English tokens' embeddings.

MATT, in the way it uses existing embeddings, is conceptually closer to FOCUS than Transtokenizers, as it models inter-token communication of original tokens, focusing predominantly on Ukrainian ones. As denoted in Appendix B, MATT quickly converges, requiring little data to recover a large part of the original model's performance. This means that small differences in the initialization (the difference between average scores for FOCUS and Transtokenizers is less than 10% in the case of Gemma 3 12B PT; see Table 1) are evened out during training. This can also be seen with WECHSEL, which performs considerably worse compared to FOCUS or Transtokenizers (Table 1), but achieves only slightly worse results after a round of MATT training, especially on higher lay-

ers (see Table 5). The final observation is presented in Table 1, where we see that even the NTP baseline starting from the Transtokenizers initialization, which is initially better, achieves similar performance to the NTP over FOCUS.

Minor efficiency differences in Table 5 are likely due to external factors such as checkpointing overhead.

**AIM on higher layers leads to better results, but saturates at around one third of the model's depth.** MATT allows selecting how deep into the model the AIM objective is applied, creating a natural trade-off between efficiency and accuracy. We evaluate different target depths and find that performance steadily improves as AIM is applied to higher layers (see Figure 5), but gains saturate once the objective reaches roughly one third of the model's total depth. In contrast, memory consumption and training time continue to grow almost linearly with the number of layers, highlighting the cost of deeper alignment.

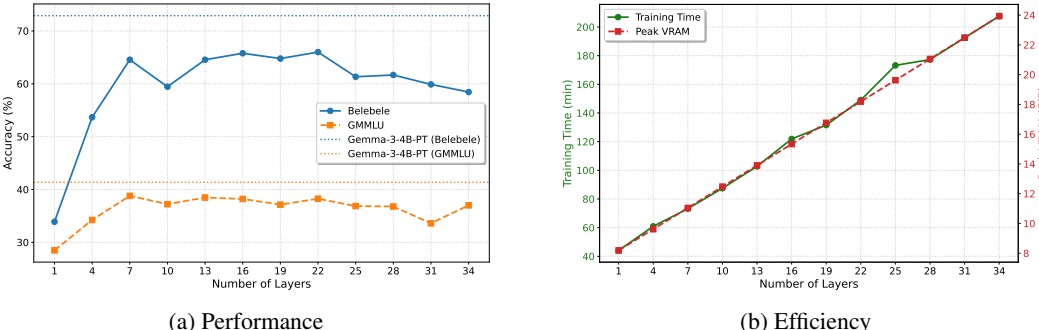

(a) Performance            (b) Efficiency

Figure 5: Effect of applying the AIM objective to different numbers of layers. The plots show the trade-off between model performance (a) and computational efficiency (b) as the application depth increases.

## D  ADDRESSING THE TIED EMBEDDINGS REQUIREMENT

The use of tied embeddings varies greatly both between the model families and model sizes. For example, Llama 3.2 1B and Llama 3.2 3B (Grattafiori et al., 2024) both utilize tied embeddings to reduce the number of parameters, whereas a larger Llama 3.1 8B does not. In contrast, the Gemma family (Team et al., 2024a;b; 2025) consistently uses tied embeddings across all sizes.

A significant amount of model pretraining research conducts very little to no experimentation on the effects of embedding tying (Jiang et al., 2023; Touvron et al., 2023; Groeneveld et al., 2024; Team, 2025). And those that do (Bai et al., 2023) offer a limited explanation of the reasons behind their choice, referring to preliminary results that are not reported in the paper.

More recent research suggests that embedding tying is more effective both from a theoretical standpoint (Bertolotti & Cazzola, 2024) and in achieving lower validation loss and better performance on downstream tasks (Allal et al., 2025). This leads us to believe that the share of models with tied embeddings may increase in the coming years, making our method even more relevant.

We conducted additional experiments on Mistral 7B v0.1 (Jiang et al., 2023), which does not tie embeddings. The results are presented in Table 6. The original Mistral's tokenizer has a vocabulary size of 32,000 and achieves a compression rate of 2.24 on Ukrainian data. We transfer Mistral to an extended vocabulary comprising over 177k tokens (a 5.5x increase, with a 4.10 compression rate). We compare the original model's performance to the FOCUS initialization, NTP optimization over both input and output embeddings, MATT with different compute budgets, and MATT combined with further NTP optimization, where we first train input embeddings using MATT, and then train only output embeddings using the NTP objective to match the budget of the NTP baseline.

MATT is unable to reach the NTP baseline (an average BLEU score of 1.55), even in a two-stage setting (with an average BLEU score of 0.63). The experiment increases the vocabulary size by more

Table 6: Benchmark results for Mistral 7B v0.1 with untied embeddings. Training time for "MATT w/ NTP (out)" reflects time spent separately on MATT and NTP on output embeddings, respectively.

| Model | Training Time | Long FLORES | WMT | XL-Sum | Avg |
|---|---|---|---|---|---|
| Mistral 7B v0.1 | - | 7.07 | 1.91 | 4.12 | 4.37 |
| FOCUS | - | 0.14 | 0.06 | 0.09 | 0.10 |
| FOCUS w/ NTP | 4h 17m | **2.06** | **0.75** | **1.85** | **1.55** |
| MATT | 2h 04m | 0.21 | 0.08 | 0.40 | 0.23 |
| MATT | 4h 08m | 0.16 | 0.09 | 0.37 | 0.21 |
| MATT | 5h 19m | 0.17 | 0.06 | 0.38 | 0.20 |
| MATT w/ NTP (out) | 2h 04m + 2h 32m | 0.54 | 0.59 | 0.76 | 0.63 |

than five times, drastically changing the model's dynamics, which additionally contributes to why training input and output embeddings jointly is of greater advantage than our two-stage approach. This also leads to a significant performance drop to the point where metrics are close to random generation, preventing a meaningful comparison of certain settings, such as MATT with different compute budgets. We leave designing another experiment with a more favorable configuration for future research.

Another potential way to handle untied embeddings is to follow Token Distillation (Dobler et al., 2025), which combines distillation on the last hidden layer with the NTP objective to optimize both input and output embeddings, albeit at the cost of higher computational requirements.

We also conducted preliminary experiments with a mapping technique that transfers input embeddings to the output embeddings space. In this setup, the output embeddings for new tokens were initialized using the mapped input embeddings after MATT fine-tuning. However, this approach underperformed compared to initializing with FOCUS and then fine-tuning only input embeddings with MATT. This can be attributed to low-capacity mapping models and requires further research.

Despite these early results, the search space remains large, and we believe that more effective strategies for untied embeddings are likely to be found with further exploration.

