# OpenReview forum: "Model-Aware Tokenizer Transfer"
_ICLR.cc/2026/Conference — Submitted to ICLR 2026_

### Official Review · Reviewer_2kWf · 2025-10-26

**Soundness:** 4
**Presentation:** 4
**Contribution:** 3
**Rating:** 8
**Confidence:** 5

**Summary:**

In this work, the authors take existings LLMs and swap their tokenizers for more efficient inference (and if desired finetuning) in a desired target language (e.g. Ukrainian). The novelty of their approach is to not only consider token embeddings on their own while reinitializing the embddings, but to also consider their perception by the model's attention layers. While considering the attention outputs does require undergoing a slight finetuning after embedding reinitialization, the efficiency of their propsed attention alignment objective is probably higher than traditional autoregressive finetuning. The authors show that their training methodology is lightweight, achieves better outcomes than embedding reinitialization alone, and bridges the gap between simpler and more advanced initializations.

**Strengths:**

* Well-explained intuition and very reasonable assumptions.
* Outstanding and well-summarized review of the state-of-the-art in tokenization trade-offs and embedding adaptation.
* Numerous baselines and ablation studies, which bring confidence in the results, and offer many potential insights.
* Very reasonable compute requirements for the achieved inference speed improvements achieved.

**Weaknesses:**

* Unlike the selected baselines, MATT involves training time on a GPU. A fairer baseline would thus consider how much a FOCUS or Transtokenizers baseline would improve after seeing ~3h of embedding finetuning on a H100 (using a more traditional autoregressive loss). After reading the paper, I assume that the proposed approach would likely perform better at a small training regime, but it might saturate sooner as the signal is only an approximation of the real objective. This would be good to objectify.
* Most LLMs do not use tied input/output embeddings, which reduce the effectiveness of the proposed method for these popular use cases. For the output embeddings, a more effective embedding transfer method and autoregresive finetuning might thus serve the use case better.

**Questions:**

* Have you considered how performance improves after a more traditional finetuning of the model using its natural autoregressive finetuning?
* Do you have an explanation for the reason why FOCUS and Transtokenizers perform about as well after MATT finetuning, but Transtokenizers perform much better alone? Are the correlations learnt by Transtokenizers' token alignment similar to the ones recovered by your approach?

---

> ### Author Response · Authors · 2025-11-25
>
> We are deeply grateful for your feedback and appreciation of our work. We also want to thank you for pointing out the key weakness of the publication - missing baselines. This is a valuable contribution to improving it, and we conducted further experiments to address these weaknesses.
>
> **W1/Q1.** We address this major weakness by conducting additional experiments, which we describe in detail in [P1](https://openreview.net/forum?id=IyV1QEc95F&noteId=dOhnCUSiHU). The results show a significant gap between the NTP baseline and MATT, both in language understanding tasks (74.01% for the best NTP baseline, 77.27% for MATT) and especially in generation (4.03 BLEU for the best NTP baseline, 6.45 for MATT).
>
> **W2.** The effectiveness of MATT is indeed limited to models with only tied embeddings. We discuss this limitation further in [P3](https://openreview.net/forum?id=IyV1QEc95F&noteId=mXqEJmKqXO).
>
> **Q2.** Transtokenizers method may have an upper hand thanks to the better utilization of English embeddings, since it learns an English-Ukrainian token-level dictionary on the parallel corpora, and utilizes it to transfer embeddings from English tokens to their Ukrainian counterparts. Whereas FOCUS uses the tokens' overlap to train a FastText model over it, and while it contains English tokens as well, there is little data in the FastText training corpus that encompasses both English and Ukrainian text in the same document. This could potentially limit the FOCUS to pay attention mostly to Ukrainian overlapped tokens, given the limited usability of English tokens’ embeddings.
>
> MATT, in the way it uses existing embeddings, is conceptually closer to FOCUS, than Transtokenizers, modeling inter-token communication of original tokens, focusing mostly on Ukrainian ones. As denoted in Appendix B, MATT quickly converges, requiring little data to recover a large part of the original model’s performance. This means that small differences in the initialization (the difference between average scores for FOCUS and Transtokenizers is less than 10%; see Table 1) are evened out during training. This can also be seen with WECHSEL, which performs considerably worse compared to FOCUS or Transtokenizers (see Table 1), but achieves only slightly worse results after a round of MATT training, especially on higher layers (see Table 5). The final observation is presented in [P1](https://openreview.net/forum?id=IyV1QEc95F&noteId=dOhnCUSiHU), where we see that even the NTP baseline starting from the Transtokenizers initialization, which is initially better, achieves similar performance to the NTP over FOCUS.

---

> > ### Comment · Reviewer_2kWf · 2025-11-26
> >
> > Thank you for comparing MATT with the CVA+NTP training baselines! The fact you both achieve better results sooner, but also likely a better plateau is a very welcome addition to the paper. I confirm my intuition that this is a paper worth publishing at a good venue.

---

### Official Review · Reviewer_GENk · 2025-10-28

**Soundness:** 1
**Presentation:** 3
**Contribution:** 2
**Rating:** 2
**Confidence:** 4

**Summary:**

This paper proposes a novel method to initialize the embeddings of newly added tokens for cross-lingual transfer effectively. The proposed method aims to incorporate higher-layer model dynamics into the initialization of new embeddings by forcing the model with the target (adapted) tokenizer to generate output embeddings similar to those produced by the source tokenizer. This is achieved with the attention influence modeling objective and a relatively small amount of training. The experimental results on both Gemma3 and Qwen3 demonstrate that the proposed approach outperforms existing initialization baselines that do not go through training.

**Strengths:**

1. This paper focuses on an important problem of how to efficiently transfer vocabulary from source to target with a particular focus on vocabulary expansion settings. While investigated by many previous studies, this paper proposes a fundamentally new approach that seeks to incorporate attention-weight information between source and target tokenizers for transfer, accounting for higher-layer model dynamics. This aspect has been neglected in previous studies and can be seen as one of the major contributions of this study in terms of originality and significance.
2. The paper is mostly well-written and easy to read up until Section 3. In particular, the intuition and motivation are pretty clear in the paper.

**Weaknesses:**

1. The fundamental limitation of this work is that the method is only applicable to input embeddings. Some frontier models, like Qwen3, do not tie weights. In that case, the effectiveness of the proposed method substantially diminishes compared to models with weight tying (Table 3).

2. The experiments and evaluation are neither extensive nor comprehensive. Specifically,
    * (i) The main adaptation experiments in 4.1 are only conducted in Ukrainian without justifications, which fails to demonstrate the applicability to different languages with different scripts.
    * (ii) all target languages in the paper are officially supported in the tested model - Qwen3 (i.e., it lacks experiments on unseen languages or unsupported languages.) While Gemma3 does not provide a list of supported languages, it is highly likely to support the target languages considering the vocabulary size and multilingual performance reported in the technical paper. Therefore, it is not clear whether the proposed method exhibits a similar advantage of better downstream performance when applied to unseen or unsupported languages.
    * (iii) The paper considers only English-to-target machine translation as a generation task, which lacks the diversity of evaluation.
    * (iv) While the paper reports compression rates as the average number of characters, it fails to report the actual inference speedups (e.g., tokens/sec used in previous related papers - See Hong et al. (2024) and Yamaguchi et al. (2024, 2025). This prevents us from evaluating the actual efficiency impact of using the proposed method.
https://arxiv.org/abs/2401.10660
https://arxiv.org/abs/2406.11477
https://arxiv.org/abs/2412.11704
    * (v) The main experiments in 4.1 and the multilingual experiments in 4.2 use different models without justifications. Also, 4.2 lacks generation evaluation. This hinders comparison between monolingual and multilingual settings.
    * (vi) The multilingual evaluation lacks in-depth analysis regarding observed differences between languages and models.

3. The comparison against baselines seems unfair, making it impossible to understand the true gain from the proposed method. While the paper argues that evaluating only the initialized model is sufficient to compare MATT with existing baselines (L477-479), MATT indeed involves training. Given that the other baselines do not include training of the base model, the paper should continually pre-train all baselines using the same compute budget to make them comparable. Moreover, one can assume that the longer the continual pre-training, the smaller the difference between initialization methods. Therefore, it is also important to demonstrate how well the performance advantage of the proposed method holds when conducting continual pre-training.

4. The discussion on the instruction-tuned model in L405-407 is entirely not reliable, given that (i) generation evaluation is conducted only on an English-to-target machine translation task and (ii) only Ukrainian is tested in the evaluation.

**Questions:**

1. The attention-related definitions in Section 3 only consider a single-head scenario. Given that almost all transformer-based models use multi-head attentions, this aspect warrants clarification (at least by a footnote).

2. How the paper chooses $n$ (i.e., how many layers are considered for the loss computation) is unclear in the main body of the paper. This should be clearly explained in the main text, as it is the core hyperparameter of the method.

3. On Weakness 1, what happens when an LM head is tuned using MATT? Does it improve performance?

4. On Weakness 2 (i), why does the paper consider only Ukrainian? The paper should expand its evaluation to different typological languages.

5. On Weakness 2 (ii), the paper should consider languages that are not supported in Qwen3. For instance, Amharic is not on the list. There are many other languages that are not on the list but are included in FLORES, Belebele, and Global MMLU.

6. On Weakness 2 (iii), the paper should consider another generation task like summarization. I believe most of the tested languages in this paper are included in XL-SUM.

7. On Weakness 2 (iv), the paper should consider including results on actual inference speedups.

8. On Weakness 2 (v), the paper should try to use the same set of tasks and models across the paper. If this is not possible, the paper should give justifications.

9. On Weakness 2 (vi), the paper should include an additional discussion on the observed performance difference between languages and models in Section 4.2.

10. On Weakness 3, the paper should conduct continual pre-training for all baselines using the same budget as MATT. Moreover, I would recommend including some full continual pre-training results for the proposed method and the best-performing baseline to examine the extent to which the advantage of the proposed method holds.

11. On Weakness 4, the paper should expand the evaluation further to make the corresponding statement reliable. If this is not possible, the discussion requires revision.

---

> ### Author Response · Authors · 2025-11-25
>
> We would like to express our gratitude to you for recognizing the novelty of the proposed method and for your extensive feedback on how the paper can be further improved.
>
> **W1/Q3.** The effectiveness of MATT is indeed limited to models with only tied embeddings. We discuss this limitation further in [P3](https://openreview.net/forum?id=IyV1QEc95F&noteId=mXqEJmKqXO).
>
> **W2 (i)/Q4.** The rise of sovereign LLMs has been a significant motivation for this research. Many of these models still suffer from inadequate tokenizer coverage, resulting in slower fine-tuning and inference. Ukrainian is one of such languages, and given the recent community discussions arguing for the need for a sovereign LLM [16], we found it to be a perfect candidate for our main experiments.
>
> To avoid limiting the evaluation to a single language, we manually select five languages that differ in resource availability and typology, yet are well-represented in benchmarks, namely English, German, Arabic, Japanese, and Swahili. We provide results for these languages in Section 4.2, although not as extensive as in our main experiments.
>
> **W2 (ii)/Q5.** Our main focus is on utilizing MATT to recover the original model’s performance after tokenizer transfer. It performs self-distillation, thus if the original model has never seen a given language (resulting in random-level performance), MATT will not be able to outperform it in a statistically significant way. This is a matter of cross-lingual transfer, which is beyond the scope of this paper.
>
> On the other hand, we can see how MATT performs when the original model produces results that are only slightly better than random, as seen in Qwen3 0.6B on Swahili benchmark subsets in Table 3, Section 4.2. Although formally the Qwen3 family supports Swahili (it is unclear whether the smallest 0.6B model was trained on it), the observed results show minimal difference compared to random-level performance. Even in this unfavorable setting, which is very similar to what it would look like for unseen/unsupported languages, MATT manages to recover most of the original model’s performance.
>
> We discuss it further in [P4](https://openreview.net/forum?id=IyV1QEc95F&noteId=Yh4QBjxX38).
>
> **W2 (iii)/Q6.** We extended the experiments in Section 4.1 with two additional generation benchmarks: WMT (another translation benchmark) and XLSum (a text summarization benchmark). We report average scores separately for language understanding and generation benchmarks. The results are presented in [P1](https://openreview.net/forum?id=IyV1QEc95F&noteId=dOhnCUSiHU).
>
> **W2 (iv)/Q7.** Since we do not introduce any changes to the architecture, the generation speed (tokens/sec) remains the same. The amount of FLOPS required to generate a single token is the same (omitting the embedding matrix extension, which can be avoided by sacrificing the coverage of the least relevant languages for a specific use case). The speed-up, however, can be measured in the time required to process the same query or generate the same text. Since tokens/sec remains constant, the speed-up is equal to the compression rate ratio (e.g., 50% for the Ukrainian), which measures the average amount of characters encoded by a single token. Given a constant-length string, doubling the compression rate results in only half the tokens needed to encode that same string compared to the original tokenization.
>
> This can be seen during evaluation. Base Gemma 3 12B PT model takes around 3,427 seconds to finish, while the assessment of the model transferred with MATT takes only 2,601 seconds, resulting in over 30% speed-up. It should be noted that we have no guarantee that both models generate the same text, and since the transferred model is still inferior to the base model, it tends to diverge more from the prompt. For example, on LongFLORES, it often fails to generate a newline after the translation is finished, which is a criterion to stop generation, and instead uses tabulation, continuing to generate another translation example. Although the translation itself is mostly accurate, this additional example negatively impacts both performance scores and generation time. Still, even in such an unfavorable setting, we see a speed increase of over 30%.
>
> **W2 (v)/Q8.** We have opted for Gemma 4B and Qwen 0.6B in the multilingual setting in order to reduce the computational cost associated with running the training and inference on a much larger set of language/model combinations.
>
> **W2 (vi)/Q9.** We will extend the discussion to address the model and language differences in the final version of the paper.

---

> > ### Author Response · Authors · 2025-11-25
> >
> > **W3/Q10.** This was a major weakness of our paper, as was outlined by other reviewers as well. We thus invested our efforts into conducting experiments with the NTP objective optimization as a second step for heuristic baselines. We present results in [P1](https://openreview.net/forum?id=IyV1QEc95F&noteId=dOhnCUSiHU) for 50%, 100% and 150% of the MATT computational budget assigned to NTP optimization. Unfortunately, we have not yet been able to investigate the full-scale continual pretraining effects due to limited computational resources.
> >
> > **W4/Q11.** We plan to remove the experiments on instruction-tuned Gemma 3 in Section 4.1, which are not comprehensive and thorough enough. They were added to support the paper’s claims, but our new experiments with the NTP objective provide a much better perspective on how MATT performs.
> >
> > **Q1.** Thank you for pointing out this oversight. We added a footnote clarifying that, although the definition is for a single-head attention, it is directly applicable to multi-head, multi-query, and grouped-query attention variants.
> >
> > **Q2.** The selection of the layer is discussed in [P2](https://openreview.net/forum?id=IyV1QEc95F&noteId=mo0gut2RbA). We will extend Section 3.5 to contain more concrete information on how $n$ is chosen.

---

### Official Review · Reviewer_4cGV · 2025-10-31

**Soundness:** 3
**Presentation:** 2
**Contribution:** 3
**Rating:** 6
**Confidence:** 3

**Summary:**

This work aims to tackle transfer of pre-trained language models on a source language to a certain target language. This is critical as the distribution of data availability is not uniform across languages and knowledge of a pre-trained model may enable more efficient transfer to obtain capable language models for target languages. To this end, the authors propose model-aware tokenizer transfer (MATT) which, contrary to existing methods, leverages the idea that a tokenizer for a target language should preserve the attention weight patterns of the source language to maximize performance. This is instantiated via Attention Influence Modeling (AIM) that aligns softmax weights between source and target tokenizers at overlapping string segments. MATT outperforms commonly used heuristics on various target languages demonstrating strong performance and improved compression rate.

**Strengths:**

- The paper is well motivated and the idea is novel to the best of my knowledge
- The results seem to be promising compared to other baselines and heuristics
- The method enables efficient transfer from a source language to a variety of target languages.

**Weaknesses:**

Generally, I am positive about the paper, however there are some clarifications required for me to fully comprehend the relevance and results reported in this work.

**Baseline comparison**

I am not convinced that Table 1 provides a fair comparison, as the authors mentioned, most of the competitors are mere heuristics that do not involve any optimization procedure whereas MATT involves an optimization process.
The comparison would be more fair if the amount of compute that was used for MATT optimization is invested in LM training for the heuristics, as this is what they usually rely on as a second step. This could be matched by the amount of compute that is needed for MATT and allocate the same amount of compute (like shown in Table 3) to tuning the embedding layer for heuristics on the language modeling task.
Eventually, this would add a new set of results that shows different compute budgets and a comparison for those of MATT to others, which would add strength and rigor to the paper.

**Clarity**

The clarity of the experimental design could be improved. For me it was sometimes not clear whether the evaluation was done in the source or in the target task and it was not explicitly stated.
For example, Belebele spans 122 languages, was the evaluation in Table 1 only done on the Ukrainian subset? Same question for GlobalMMLU and LongFLORES.

In Table 3 there are some languages for which the base model is still better even after tokenizer transfer, like arabic or japanese. As far as I understand this provides negative results -- why would you do tokenizer transfer if the base model performs better on certain langauges anyways? Maybe I misunderstood this part, but I believe it is central to clarify the relevance of this work. If it is indeed a negative result, is there a way to find a proxy for when tokenizer transfer would work and when it doesn't?

Finally, it is not entirely clear to me whether after tokenizer transfer during inference one still needs to sort of switch between tokenizers depending on the language one wants to predict for, i.e. do I need to know during inference whether I evaluate on a target language or not? In my view this knowledge is required, but from the way certain parts of the text are written it sounds like it is not, can the authors please elaborate on that?

**Significance of results**

The performance difference between different components of MATT in Table 1 and Table 3 is sometimes marginal, therefore it is not clear what setup generally works best or should be used by default. I recommend to add error bars at least for the ablation studies to see whether different setups exhibit statistically significant performance.


**Limitations**

If I understood correctly, another limitation that wasn't mentioned is that for collecting the targets for the optimization process you need an extra forward pass, once with the original tokenizer and once with the external tokenizer which is then used for backpropagation to obtain gradients for the embedding weights.

**Questions:**

- Line 337: What does it mean to merge tokenizers? This is a bit confusing to me as it sounds that the training process is then agnostic to tokenizers, but for each input you still need to segment based on each tokenizers separately to obtain inputs to your loss function, right?
- Line 361: Why is the MSE loss chosen to be on the 12th layer? I see in Table 5 all layers were tested, but were there also other single layers tested except layer 12? Could it be that some other layers provide more useful information? It might be interesting to look at earlier or intermediate layers as well.

---

> ### Author Response · Authors · 2025-11-25
>
> We greatly appreciate your input on our work and recognition of its strengths. We apologize for the confusion we may have introduced in the method’s description, which led to unclear motivation and purpose of MATT. Therefore, we address the outlined weaknesses and clarify the experiments below.
>
> **Weakness: Baseline comparison**
>
> This was a major weakness of our paper, as was outlined by other reviewers as well. We thus invested our efforts into conducting experiments with the NTP objective optimization as a second step for heuristic baselines. We present results in [P1](https://openreview.net/forum?id=IyV1QEc95F&noteId=dOhnCUSiHU) for 50%, 100% and 150% of the MATT computational budget assigned to NTP optimization.
>
> **Weakness: Clarity**
>
> We apologize for any misunderstanding regarding our use of the term “tokenizer transfer”. Our intention was not to imply cross-lingual transfer. Tokenizer transfer does not necessarily aim to introduce a new language to the model; rather, it adapts the tokenization procedure for already supported languages. The primary motivation is improved training and inference efficiency: by representing the same text with fewer tokens, we reduce overall processing and generation time.
>
> In Section 4.1, we transfer the original Gemma tokenizer to a modified Ukrainian-specific tokenizer, which reduces the number of tokens required for the same text by more than 30%. This directly results in a ~50% inference speedup, as stated on line 346 of the original submission.
>
> As noted on lines 48–50, tokenizer transfer is not expected to introduce new knowledge into the model, especially when distillation from the original model is involved. Therefore, our goal is to match the performance of the original model. Deviations above that baseline are likely due to statistical noise, which explains why most values in Table 3 are slightly below those of the base model.
>
> The experiments in Section 4.1 evaluate only Ukrainian subsets of Belebele and Global MMLU, as well as the English-to-Ukrainian portion of LongFLORES, to assess generation quality. All baselines, except the original models, use the same Ukrainian-specific tokenizer for fairness.
>
> Section 4.2 follows a similar setup, but here we train a single tokenizer designed to support multiple languages more effectively, as shown in Table 2. We evaluate the Arabic, German, English, Japanese, and Swahili subsets of Belebele, MMMLU, and Global MMLU to measure the resulting multilingual performance.
>
> One of the use cases for MATT is aimed at the scenario of tokenizer adaptation for languages already supported by a given LLM, but producing multiple tokens even for popular words. In such a case, the tokenizer can be adapted by extending the dictionary with BPE training on the target language, then initializing the new token embeddings with methods such as Transtokenizers or FOCUS, and following a MATT training. This scenario will typically yield thousands or tens of thousands of new tokens. In our experiment in Section 4.1, we apply it to 80 thousand new tokens.
>
> **Weakness: Significance of results**
>
> We plan to remove the experiments on instruction-tuned Gemma 3 in Section 4.1, which are not comprehensive and thorough enough. They were added to support the paper’s claims, but our new experiments with the NTP objective provide a much better perspective on how MATT performs.
>
> As for Table 3, we will extend a discussion on the differences between different MATT settings.
>
> **Weakness: Limitations**
>
> Yes, that is absolutely correct. We will update the limitations accordingly.
>
>
> **Q1.** We experiment with BPE tokenizers, which form new tokens by merging already existing ones. To extend the base tokenizer, we first train a separate tokenizer on the target language with the same hyperparameters as the original one. Then we identify all tokens that are absent from the base vocabulary. For each such token, we search for all merge operations, combinations of already available subwords, that can produce this token. We then add the new token and all corresponding merges to the base tokenizer. This process is iterative: when processing later tokens, we also consider the tokens and merges introduced in earlier steps. In this way, the base tokenizer is expanded to enhance its language coverage while maintaining consistency with its original subword structure.
>
> **Q2.** We provide arguments for this decision in [P2](https://openreview.net/forum?id=IyV1QEc95F&noteId=mo0gut2RbA).

---

### Official Review · Reviewer_emKC · 2025-10-31

**Soundness:** 2
**Presentation:** 3
**Contribution:** 2
**Rating:** 2
**Confidence:** 5

**Summary:**

The paper proposes a method to transfer the tokenizer of a pretrained LLM to a new tokenizer. It initializes embeddings for new tokens by optimizing a distillation objective based on matching segments of hidden states. The method is evaluated in language adaptation settings and compared against several other “heuristic” embedding initialization methods.

**Strengths:**

**S1:** The proposed method significantly outperforms (non-learning based) heuristic embedding initialization methods in the reported evaluations.

**S2:** The method’s intuition on distilling inter-token attention patterns is well-motivated.

**S3:** The paper is well-written and easy to follow, with good figures to explain the proposed method.

**Weaknesses:**

**W1:** There are several important baselines missing, which are required to properly contextualize the strong results of the proposed method. In general, the current experimental evaluation is insufficient and forgoes comparison against stronger (non-heuristic) embedding initialization baselines. **It is not surprising** that directly optimizing new embeddings on 240 million (!) tokens outperforms heuristic initializations based on some mappings between embedding spaces without any further training. Therefore, I recommend adding comparisons against the following baselines in a future revision to back up the strong claims of the paper:
- **W1.1:** The method compares only against “heuristic” embedding initialization methods which are not learning based. Apart from ZeTT (which amortizes the learning during its hyper-network pretraining), one crucial baseline is learning the new embeddings via the *classic next-token prediction objective* using the same setup as in the experiments for the proposed AIM objective (e.g. same data, computational budget, ..). **This baseline is necessary** and should have been provided in all experiments to evaluate the usefulness of the proposed (more complicated) objective vs. a simpler existing learning objective.
 - **W1.2:** As ZeTT (https://arxiv.org/abs/2405.07883) is very competitive, a comparison of the proposed method to ZeTT is desirable. Can the direct optimization of the embeddings via MATT beat the embeddings produced by ZeTT?
- **W1.3:** Another non-heuristic baseline is the method proposed in https://arxiv.org/abs/2410.05864, which extracts new token representations from intermediate layers and maps them back to the embedding spaces. This method also outperforms heuristic-based initialization methods in their evaluations.

**W2:** In l. 81ff:
> Our contributions are: [...] Model-Aware Tokenizer Transfer (MATT): an efficient tokenizer-transfer method that exploits model dynamics instead of relying solely on semantic relationships, achieving state-of-the-art results with substantially lower computational cost than language modeling objectives.

 I could not find any comparison to language modeling objectives (i.e. next-token prediction with cross-entropy?) in the paper, could you point me to the correct Figure/Table?

**W3:** In l. 137ff you write:
> Other work (Abagyan et al., 2025) shows that periodically resetting embeddings during pretraining makes models more robust to them, reducing the effort needed to learn new tokens afterwards.

but I cannot find support for this claim anywhere in the paper by Abagyan et al., 2025. Could you elaborate?




**Score summary:** I recommend to **`reject`** the paper in its current state due to (1) missing baselines which are necessary to evaluate the strength of the proposed method compared to prior work / simpler methods (see W1), (2) claims for which I could not find support for in the paper/cited references (W2 & W3) and (3) questions regarding novelty compared to insufficiently discussed prior work (Q2). If these points are sufficiently addressed, I am open to raising my score.

**Questions:**

- **Q1:** The training loss only optimizes the new embeddings in their role as input embeddings, as you discuss. How does this influence the model’s ability to actually generate the new tokens (in the case of tied embeddings)?
- **Q2:** The proposed method’s intuition, setup, and loss objective seem (very) similar to the one proposed in https://arxiv.org/abs/2505.20133, could you discuss the differences of your work?
- **Q3:** What target layer for the AIM/AIM* objectives is used in the main experiments?
- **Q4:** How well does MATT work with varying degrees of presence of the target language(s) in the models original pretraining data? Does the model need to have already been trained on target languages?

**(nit):** this does not influence my review of the paper, but could you explain the difference between the proposed AIM objective and the MATT method? Is MATT just the application of the AIM (or AIM*) objective?

---

> ### Author Response · Authors · 2025-11-25
>
> We would like to thank you for your thorough review, for outlining strengths, and especially for highlighting weaknesses. This has been of immense help in identifying and addressing key areas for improvement.
>
> **W1.** We fully agree with the reviewer that the paper should have provided the NTP baseline with a similar compute budget. We address this in [P1](https://openreview.net/forum?id=IyV1QEc95F&noteId=dOhnCUSiHU).
>
> **W2.** This claim was indeed unsupported due to the lack of a language modeling baseline, but this was addressed with the new experiments described in [P1](https://openreview.net/forum?id=IyV1QEc95F&noteId=dOhnCUSiHU).
>
> **W3.** Thank you for spotting this mistake. You are correct. We wanted to cite Chen et al. 2023, “Improving Language Plasticity via Pretraining with Active Forgetting” (https://arxiv.org/pdf/2307.01163). Still, we made an oversight when collecting BibTeX citations, and instead cited Abagyan et al. 2025, “One Tokenizer To Rule Them All: Emergent Language Plasticity via Multilingual Tokenizers” (https://arxiv.org/pdf/2506.10766), most probably due to searching for the citation using the “Language Plasticity” keyword.
>
> **Q1.** In our preliminary experiments, we find that to fully utilize the generation of newly added tokens, further short NTP training is needed. After such training, we test the generation on a few Ukrainian-centric question-answering and translation prompts. Newly added tokens account for 60-80% of the generated responses.
>
> **Q2.** We thoroughly analyzed the Token Distillation paper, and while the intuition is quite similar, our methods have different and, to some extent, complementary approaches.
>
> 1. Token Distillation follows a commonly used hidden state distillation approach, which is possible because both the teacher and the student originate from the same model, thereby sharing the latent space. Distilling over hidden states provides the student model with a much richer signal and has long been known to improve the performance of the student model [17].
> 2. The main modification, which enables the application of Token Distillation in the cross-tokenizer setting, involves aligning tokens corresponding to the same words and using only the last token of each word. This has a clear intuition: the hidden state of the last token of each word is used to generate the first token of the following word, the very same idea we came up with when defining the query state of the segment to be equal to the query state of its last segment.
> 3. The key difference between MATT and Token Distillation is that, while following a similar intuition, MATT disentangles the effect the context has on the generation of the next token by exploiting inter-token patterns in the attention layer. This establishes a novel approach to distillation.
> 4. Token Distillation distills hidden states from the last layer, which requires almost a full pass through the model. MATT, on the other hand, is applied to one of the intermediate layers, reducing the overall computational cost of training.
> 5. Token Distillation employs retrieving relevant contexts for new tokens, thereby avoiding the unnecessary computations on documents that do not contain any new tokens. This is especially valuable when conducting small-scale experiments, as it introduces a limited amount of tokens. We considered adopting a similar approach to data selection, but it was postponed due to higher-priority goals.
> 6. According to the experiments conducted in the Token Distillation paper, it is unclear whether its effectiveness holds when introducing tens of thousands of new tokens, as we did in our experiments, adding over 80,000 new tokens in Section 4.1. The experiments in the Token Distillation paper add only a few thousand tokens, and the random embedding initialization already yields high benchmark results, due to its limited effect on the model.
> 7. Token Distillation is only defined for vocabulary extension, allowing us to add new tokens that are combinations of existing ones. MATT, on the other hand, uses offset-based matching, which enables us to match many-to-many tokens, thereby being more flexible in the choice of the target tokenizer.
>
> **Q3.** We use the 12th layer out of 34. We discuss this decision further in [P2](https://openreview.net/forum?id=IyV1QEc95F&noteId=mo0gut2RbA).
>
> **Q4.** Since other reviewers also wondered about the performance of MATT on unseen languages, we discuss it in [P4](https://openreview.net/forum?id=IyV1QEc95F&noteId=Yh4QBjxX38).
>
> **(nit).** Yes, you are correct. MATT is a method for tokenizer transfer that utilizes the AIM/AIM* objectives. We differentiate between those because we see other potential applications of AIM in distillation methods, not necessarily only tokenizer transfer. However, we leave this for future research.

---

### Author Response · Authors · 2025-11-25
**General Response**

We want to thank all reviewers for their insightful comments. We have taken into consideration as much of the helpful feedback as possible, but some issues were left unresolved due to time constraints. We apologize for the late response. We dedicated a lot of time to addressing as many suggestions as possible. We provide all results in this response and will incorporate them into the paper version the following day.

The response is organized as follows. In the main section, we address the problems relevant to multiple reviewers. These problems are numbered for easy reference. We number and refer to them with P1, P2, etc. We refer to specific weaknesses and questions in responses to the specific reviewer by W1, W2, … (weaknesses) and Q1, Q2, … (questions).

Regarding the references, we use consistent paper citations across the whole response. The references are listed at the end of the main section.

---

## References

[1] Llama 3.2 1B (https://huggingface.co/meta-llama/Llama-3.2-1B)

[2] Llama 3.2 3B (https://huggingface.co/meta-llama/Llama-3.2-3B)

[3] Llama 3.1 8B (https://huggingface.co/meta-llama/Llama-3.1-8B)

[4] Mistral 7B (https://arxiv.org/pdf/2310.06825)

[5] LLaMA: Open and Efficient Foundation Language Models (https://arxiv.org/pdf/2302.13971)

[6] Qwen3 Technical Report (https://arxiv.org/pdf/2505.09388)

[7] OLMo: Accelerating the Science of Language Models (https://arxiv.org/pdf/2402.00838)

[8] Qwen Technical Report (https://arxiv.org/pdf/2309.16609)

[9] By Tying Embeddings You Are Assuming the Distributional Hypothesis (https://openreview.net/pdf?id=yyYMAprcAR)

[10] The Smol Training Playbook: The Secrets to Building World-Class LLMs. (https://huggingface.co/spaces/HuggingFaceTB/smol-training-playbook)

[11] Mistral 7B v0.1 (https://huggingface.co/mistralai/Mistral-7B-v0.1)

[12] Token Distillation: Attention-aware Input Embeddings For New Tokens (https://arxiv.org/pdf/2505.20133)

[13] From Tokens to Words: On the Inner Lexicon of LLMs (https://arxiv.org/pdf/2410.05864v1)

[14] Language Contamination Helps Explain the Cross-lingual Capabilities of English Pretrained Models (https://arxiv.org/pdf/2204.08110)

[15] A simple, fast, and effective reparameterization of IBM Model 2 (https://aclanthology.org/N13-1073.pdf)

[16] Why Does Ukraine Need Its Own LLM? (https://digitalstate.gov.ua/news/govtech/navishcho-ukrayini-vlasna-llm)

[17] DistilBERT, a distilled version of BERT: smaller, faster, cheaper and lighter (https://arxiv.org/pdf/1910.01108)

---

> ### Author Response · Authors · 2025-11-25
> **P1. Evaluation baselines**
>
> We fully recognize the introduced unfairness in the evaluation, as it does not include a dedicated baseline that optimizes over the embedding layers with a similar compute budget to our method. This is, indeed, a serious oversight that limits the robustness and rigor of the paper. It has become our priority to conduct experiments with Next Token Prediction (NTP) training, starting from the FOCUS and Transtokenizers initializations, which have shown the best performance among the tokenizer transfer methods in our main experiments (see Table 1).
>
> | Model | Steps | Training Time | Belebele | Global MMLU | Long FLORES | WMT | XLSum | Avg NLU | Avg Gen |
> |-------|:-----:|:-------------:|:--------:|:-----------:|:-----------:|:---:|:-----:|:-------:|:----:|
> | Gemma 3 12B PT | - | - | 89.33 | 67.03 | 14.36 | 3.52 | 6.52 | 78.18 | 8.13 |
> | FOCUS | - | - | 48.78 | 37.14 | 1.01 | 0.88 | 0.20 | 42.96 | 0.70 |
> | FOCUS+NTP | 5k | 3.5h | 85.44 | 57.38 | 3.51 | 2.13 | 4.32 | 71.41 | 3.32 |
> | FOCUS+NTP | 10k | 7h | 87.00 | 60.55 | 4.32 | 2.51 | 5.04 | 73.78 | 3.96 |
> | FOCUS+NTP | 15k | 10.5h | 87.44 | 60.57 | 4.34 | 2.60 | 5.16 | 74.01 | 4.03 |
> | Transtokenizers | - | - | 61.89 | 46.03 | 0.04 | 0.09 | 0.02 | 53.96 | 0.05 |
> | Transtokenizers+NTP | 5k | 3.5h | 82.44 | 59.02 | 3.64 | 0.88 | 4.06 | 70.73 | 2.86 |
> | Transtokenizers+NTP | 10k | 7h | 85.22 | 59.83 | 4.63 | 0.95 | 4.80 | 72.53 | 3.46 |
> | Transtokenizers+NTP | 15k | 10.5h | 85.67 | 59.38 | 5.13 | 0.96 | 4.80 | 72.53 | 3.63 |
> | MATT | 257k | 7h | **89.56** | **64.98** | **8.70** | **4.71** | **5.95** | **77.27** | **6.45** |
>
> Furthermore, we report NTP results for 50%, 100%, and 150% of MATT’s compute budget in the table above. The most interesting is the minimal improvements achieved by NTP when comparing 100% and 150% compute budgets (7h and 10.5h of training time, respectively). The NTP training quickly saturates, and MATT’s performance does not seem to be achievable within a reasonable budget. We do not continue with further training due to the limited computational budget.
>
> As for the MATT, it also saturates quickly, but at a higher performance level. We speculate that further improvements are difficult to achieve with mere embedding training, and thus require unfreezing the model’s layers. This is largely due to the number of newly introduced tokens (over 80,000), which inevitably changes the model’s dynamics, and would benefit from full fine-tuning. Unfortunately, we have not yet experimented with continual pretraining after MATT due to limited computational budget.
>
> We further enhance the main results by adding two benchmarks for generation: WMT for translation, and XLSum for summarization. All generation benchmarks use the BLEU score as their main metric. Thus, we additionally report separate averages across Natural Language Understanding (NLU) and Generation benchmarks.
>
> We plan to add NTP baselines for WECHSEL and TokAlign in the camera-ready revision. However, we do not expect them to surpass those for FOCUS and Transtokenizers, given the significant differences reported in Table 1 of the submission.
>
> ZeTT results are not reported because there is no pretrained hypernetwork for Gemma 3. We made a significant effort to train it ourselves, but given that the original implementation is in JAX/Flax (https://github.com/bminixhofer/zett), which are already deprecated in Transformers (https://github.com/huggingface/transformers/pull/38758), we were unable to adapt the code to the Gemma 3 model, despite our best efforts.
>
> Another approach we discussed was to reimplement ZeTT in PyTorch. However, given the limited time for discussion, we did not take this path.

---

> > ### Author Response · Authors · 2025-11-25
> > **P2. Choosing the target layer for AIM**
> >
> > We have experimented with applying MATT to different layers in one of our ablation studies, as shown in Appendix C, Figure 5a. We find that choosing higher layers leads to better final performance up to around one-fourth of all layers. This can be attributed to the detokenization process [13], which takes place in early and middle layers and forms coherent word-level representations from input tokens. Additionally, selecting one of the last layers also results in a slight performance degradation, which is consistent with findings in Token Distillation [12].
> >
> > This leaves us with the middle layers, which exhibit similar performance, establishing a plateau. Considering Figure 5b, which shows that selecting a higher layer as a target increases both training time and memory requirements linearly, we pick one of the first layers on the plateau to achieve the best performance with minimal training time. In the case of Gemma 3 12B PT, we have picked the 12th layer.

---

> ### Author Response · Authors · 2025-11-25
> **P3. Untied Embeddings Limitation**
>
> The use of tied embeddings varies greatly both between the model families (Gemma, Llama, Qwen) and model sizes. For example, Llama 3.2 1B [1] and Llama 3.2 3B [2] both utilize tied embeddings to reduce the number of parameters, whereas a larger Llama 3.1 8B [3] does not. In contrast, the Gemma family consistently uses tied embeddings across all sizes.
>
> A significant amount of model pretraining research conducts very little to no experimentation on the effects of embedding tying [4,5,6,7]. And those that do [8] offer a limited explanation of the reasons behind their choice, referring to preliminary results that are not reported in the paper.
>
> More recent research suggests that embedding tying is more effective both from a theoretical standpoint [9] and in achieving lower loss and better performance on downstream tasks [10]. This leads us to believe that the share of models with tied embeddings may increase in the coming years, making our method even more relevant.
>
> We conducted additional experiments on Mistral 7B v0.1 [11], which does not tie embeddings. The results are presented in the table below. The original Mistral’s tokenizer has the vocabulary size of 32k, and achieves 2.24 compression rate on Ukrainian data. We transfer Mistral to an extended vocabulary comprising over 177k tokens (over 5.5x increase, 4.10 compression rate). We compare the original model’s performance to the FOCUS initialization, NTP optimization over both input and output embeddings for 15k steps, MATT with different compute budgets, and MATT combined with further NTP optimization, where we (1) train input embeddings using MATT, and then (2) train only output embeddings using the NTP objective to match the budget of the NTP baseline.
>
> | Model | Steps | Training Time | Long FLORES | WMT | XLSum | Avg |
> |-------|:-----:|:-------------:|:-----------:|:---:|:-----:|:---:|
> | Mistral 7B v0.1 | - | - | 7.07 | 1.91 | 4.12 | 4.37 |
> | FOCUS | - | - | 0.14 | 0.06 | 0.09 | 0.10 |
> | FOCUS+NTP | 15k | 257m | **2.06** | **0.75** | **1.85** | **1.55** |
> | MATT | 100k | 124m | 0.21 | 0.08 | 0.40 | 0.23 |
> | MATT | 200k | 248m | 0.16 | 0.09 | 0.37 | 0.21 |
> | MATT | 257k | 319m | 0.17 | 0.06 | 0.38 | 0.20 |
> | MATT+NTP(out) | 100k+9k | 124m+152m | 0.54 | 0.59 | 0.76 | 0.63 |
>
> MATT is unable to reach the NTP baseline (an average BLEU score of 1.55), even in a two-stage setting (with an average BLEU score of 0.63). The experiment increases the vocabulary size by more than five times, drastically changing the model’s dynamics, which additionally contributes to why training input and output embeddings jointly is of greater advantage than our two-stage approach.
>
> Another potential way to handle untied embeddings is to follow Token Distillation [12], which combines distillation on the last hidden layer with the NTP objective to optimize both input and output embeddings, albeit at the cost of higher computational requirements.
>
> We will continue to experiment with ideas that would allow us to extend MATT’s applicability to models with untied embeddings.

---

> > ### Author Response · Authors · 2025-11-25
> > **P4. Performance on unseen languages**
> >
> > Since MATT performs a self-distillation, its performance depends on the original model’s performance, which, in turn, is highly correlated with the presence of target languages in its pretraining data [14]. This means that we cannot expect to achieve significantly better performance in a target language compared to the original model. However, it is worth noting that the commonly used definition of a certain model supporting a certain language is often defined by its creators to signify whether the model outperforms a specific metric's value threshold, rather than the presence of training data for the given language.
> >
> > Extending MATT to cross-lingual transfer would most probably require introducing a new alignment strategy, which would match words from sentences in different languages (we could utilize parallel corpora for this and some alignment methods, like FastAlign [15], similarly to Transtokenizers). However, it is unclear whether modeling inter-token attention patterns in one language is beneficial to the model’s performance in another. This definitely requires conducting many more experiments, which is beyond the scope of the paper, even though it sounds like a promising idea for future research.

---

### Author Response · Authors · 2025-12-02
**Official Summary by Authors**

Dear Reviewers, Area Chairs, and Senior Area Chairs,

In light of the measures taken in response to the recent incident, we would like to provide you with a brief summary of the review process that was conducted, which should help you navigate it more easily.

We also want to thank reviewers for their thorough examination of our work and thoughtful comments and suggestions, which allowed us to improve the paper’s rigor, clarity, and insightfulness. Furthermore, we would like to express our gratitude to Area Chairs, Senior Area Chairs, and everyone who has put in a great deal of effort to navigate the current situation and ensure fair and unbiased decision-making.

Our responses to reviewers’ concerns and suggestions follow a specific format for clarity and navigability. We describe this format in the [General Response](https://openreview.net/forum?id=IyV1QEc95F&noteId=MfITD3HfuV).

---

References

[1] Sovereign Large Language Models: Advantages, Strategy and Regulations (https://arxiv.org/abs/2503.04745)

---

> ### Author Response · Authors · 2025-12-02
> **Motivation, Novelty, Strengths**
>
> With the greater adoption of LLMs by the public, businesses, and governments, a high demand has arisen for language- and domain-adapted local models that are both powerful and efficient. This translated into multiple endeavors to meet this demand: BgGPT (Bulgaria), Bielik, PLLuM (Poland), Lapa LLM, MamayLM (Ukraine), EuroLLM (Europe), BharatGen (India), etc. Sovereign LLMs have a wide range of applications, from education to national defense, and already catalyze numerous processes [1].
>
> The cost of training and serving LLMs is higher for languages and domains that are underrepresented in the tokenizer’s vocabulary. We mitigate this problem by introducing Model-Aware Tokenizer Transfer, a novel self-distillation method that models the inter-token communication happening in attention layers of the original model. By comparing it to other heuristics- and optimization-based methods, we demonstrate that our method enables both a faster and higher recovery of the original model’s performance with the new tokenizer.
>
> Most reviewers outline the well-explained motivation and novelty of the proposed method. They commend the writing and presentation.

---

> > ### Author Response · Authors · 2025-12-02
> > **Primary Reviewers’ Concerns**
> >
> > The main weakness that every reviewer emphasized was the lack of optimization-based baselines in our evaluation, which led to unfair comparison, since MATT involves training over embeddings. We made it our priority to provide such baselines. The last revision of our paper includes multiple NTP baselines with different initializations and various compute budgets, all of which still underperform compared to our method. We address this more extensively in [P1](https://openreview.net/forum?id=IyV1QEc95F&noteId=dOhnCUSiHU).
> >
> > Our work lacked a clear justification for some decisions made during experiments, such as the choice of certain hyperparameters or the languages used for evaluation. We added these clarifications to the latest revision (mentioned in [P2](https://openreview.net/forum?id=IyV1QEc95F&noteId=mo0gut2RbA),in [W2 (i)/Q4](https://openreview.net/forum?id=IyV1QEc95F&noteId=8glqoY6zCh), [W2 (ii)/Q5](https://openreview.net/forum?id=IyV1QEc95F&noteId=8glqoY6zCh), [W2 (vi)/Q9](https://openreview.net/forum?id=IyV1QEc95F&noteId=8glqoY6zCh), [Q1](https://openreview.net/forum?id=IyV1QEc95F&noteId=Xu9kiePBib), [Q2](https://openreview.net/forum?id=IyV1QEc95F&noteId=Xu9kiePBib) by reviewer GENk and in [Weakness: Limitations](https://openreview.net/forum?id=IyV1QEc95F&noteId=136l1qzYvQ) by reviewer 4cGV), and extended our discussion of the experiment results.
> >
> > We acknowledge that the reduced effectiveness of our method on models with untied embeddings is a limitation. We performed additional experiments, as described in [P3](https://openreview.net/forum?id=IyV1QEc95F&noteId=mXqEJmKqXO), and added the results to Appendix D. While these results confirm the superiority of the NTP baseline in this specific setting, we argue that MATT’s applicability remains high and is poised to grow in significance. Finally, we outline several potential research directions to address this limitation in future work.
> >
> > Many other clarifications and answers are spread throughout our responses.

---

### Meta-Review · Area_Chair_4DBF · 2026-01-02

**Summary:**

The reviewers raised concerns about the limited applicability of the proposed method due to its reliance on tied input–output embeddings. They also noted that the experimental evaluation in the original submission was insufficient.

**Reviewer Concerns:**

Some key concerns remain outstanding. For example, the method’s strong reliance on tied input–output embeddings substantially limits its applicability to modern large-scale models with untied embeddings, which constrains the overall impact of the work. Also, although the rebuttal adds useful NTP baselines, the original evaluation still lacks comparisons against several critical learning-based baselines, e.g., ZeTT and other recent intermediate-layer approaches.

**Reviewer Scores:**

I believe the reviewers would have modestly increased their scores in light of the rebuttal, moving from a clear reject to a borderline reject. However, the remaining concerns would likely prevent a shift toward acceptance.

---

### Decision · Program_Chairs · 2026-01-26

Reject